

# Comparison of 4-Dimensional Variational and Ensemble Optimal Interpolation data assimilation systems using a Regional Ocean Modelling System (v3.4) configuration of the eddy-dominated East Australian Current System

Colette Kerry[1], Moninya Roughan[1], Shane Keating[2], David Gwyther[1,3], Gary Brassington[4], Adil Siripatana[1,6], and Joao Marcos A. C. Souza[5]

[1]Coastal and Regional Oceanography Lab, School of Biological, Earth and Environmental Sciences, UNSW Sydney Australia, Sydney, NSW, Australia 2052.
[2]School of Mathematics and Statistics, UNSW Sydney Australia, Sydney, NSW, Australia 2052.
[3]School of Earth and Environmental Sciences, University of Queensland, Brisbane, Australia.
[4]The Centre for Australian Weather and Climate Research, Bureau of Meteorology, Melbourne, Australia.
[5]Meteorological Service of New Zealand, MetOcean Division, Raglan, New Zealand.
[6]AI and Computer Engineering, CMKL University, Thailand.

**Correspondence:** Colette Kerry (c.kerry@unsw.edu.au)

**Abstract.** Ocean models must be regularly updated through the assimilation of observations (data assimilation) in order to correctly represent the timing and locations of eddies. Since initial conditions play an important role in the quality of short-term ocean forecasts, an effective data assimilation scheme to produce accurate state estimates is key to improving prediction. Western Boundary Current regions, such as the East Australia Current system, are highly variable regions making them par-

ticularly challenging to model and predict. This study assesses the performance of two ocean data assimilation systems in the East Australian Current system over a two year period. We compare the time-dependent 4-Dimensional Variational (4D-Var) data assimilation system with the more computationally-efficient, time-independent Ensemble Optimal Interpolation (EnOI) system, across a common modelling and observational framework. Both systems assimilate the same observations including: satellite-derived sea-surface height, sea-surface temperature, vertical profiles of temperature and salinity (from Argo floats),

and temperature profiles from eXpendable Bathy-Thermographs. We analyse both systems' performance against independent data that is withheld allowing a thorough analysis of system performance. The 4D-Var system is 25 times more expensive but outperforms the EnOI system against both assimilated and independent observations at the surface and subsurface. For forecast horizons of 5-days Root-Mean-Squared forecast errors are 20-60% higher for the EnOI system compared to the 4D-Var system. The 4D-Var system, which assimilates observations over 5-day windows, provides a smoother transition from the end of the

forecast to the subsequent analysis field. The EnOI system displays elevated low frequency ($>1$ day), surface intensified variability in temperature, and elevated kinetic energy at length scales less than 100km at the beginning of the forecast windows. The 4D-Var system displays elevated energy in the near-inertial range throughout the water column, with the wavenumber kinetic energy spectra remaining unchanged upon assimilation. Overall, this comparison shows quantitatively that the 4D-Var system results in improved predictability as the analysis provides a smoother and more dynamically-balanced fit between





the observations and the model's time-evolving flow. This advocates the use of advanced, time-dependent data assimilation methods, particularly for highly variable oceanic regions, and motivates future work into further improving data assimilation schemes.

**Keywords:** East Australian Current (EAC), Four-Dimensional Variational (4D-Var) Data Assimilation, Ensemble Optimal Interpolation (EnOI), mesoscale, prediction, Western Boundary Current, ROMS

**Key Points:**

1. The predictive performances of two ocean data assimilation systems (EnOI and 4D-Var) are assessed in a ROMS configuration of the East Australian Current over 5-day forecast horizons.

2. The forecast skill of the 4D-Var system surpasses the EnOI system against both assimilated and independent observations
at the surface and subsurface.

3. The EnOI system has greater analysis increments, elevated low-frequency ($>1$ day) surface-intensified variability in temperature, and elevated kinetic energy at length scales less than 100km at the beginning of the forecast windows.

4. The dynamically-balanced 4D-Var system displays elevated energy in the near-inertial range throughout the water column, with the wavenumber kinetic energy spectra remaining unchanged upon assimilation.

## 1   Introduction

Data assimilation (DA), the combination of numerical modelling and observations, is essential to produce accurate forecasts of the atmosphere or ocean circulation. The goal of any DA scheme is to combine observations and a numerical model such that the result is a better estimate of the ocean circulation than either alone. Observations provide sparse data points while the model
provides context. Since initial conditions play an important role in forecast quality, accurate and dynamically consistent state estimates are key to improving prediction. This study focuses on the comparison of two DA techniques applied to forecasting the ocean mesoscale circulation in a highly dynamic oceanic region.

Mesoscale eddies exist throughout the global ocean and contain more than half of the kinetic energy of the ocean circulation. Western Boundary Current (WBC) regions are hot-spots of high eddy variability as eddies emerge due to instabilities in the
strong boundary current flow. The high mesoscale eddy variability (Stammer, 1997; Mata et al., 2000) and the complexities of eddy shedding processes and evolution (Mata et al., 2006; Bull et al., 2017) make WBCs challenging to model and predict (Feron, 1995; Imawaki et al., 2013; Roughan et al., 2017). Due to the chaotic nature of the mesoscale circulation, ocean models must be regularly updated through the assimilation of observations in order to correctly represent the timing and locations of



eddies (e.g. Kerry et al., 2016; Li and Roughan, 2023) and accurate forecasts of eddies as they shed, evolve and interact in
WBC regions are lacking.

The East Australian Current (EAC), the WBC of the South Pacific subtropical gyre (Figure 1a), and its associated eddies
dominate the circulation along the southeastern coast of Australia. The southward-flowing current is most coherent off 27° S
(Sloyan et al., 2016) and intensifies at around 31° S (Kerry and Roughan, 2020). The current typically separates from the coast
between 31° S and 32.5° S  (Cetina Heredia et al., 2014) and turns eastward to form the EAC eastern extension, shedding large
warm-core eddies in the Tasman Sea (Oke and Middleton, 2000; Cetina Heredia et al., 2014; Oke et al., 2019). In the EAC,
eddies can directly influence shelf circulation (Schaeffer et al., 2014; Schaeffer and Roughan, 2015; Malan et al., 2023) and
often intensify as the jet separates from the coast. After shedding, eddies propagate and evolve (Pilo et al., 2015b, a) and can
display a complex vertical structure including tilting and stacking (Oke and Griffin, 2011; Macdonald et al., 2013; Roughan
et al., 2017; Pilo et al., 2018). As such, the EAC is a challenging region to predict and provides an ideal test-bed for comparison
of DA methods.

There are various DA techniques, by which a model estimate of the ocean state can be combined with ocean observations,
that vary in complexity. Simpler, computationally efficient, time-independent methods such as 3-Dimensional Variational Data
Assimilation (3D-Var) and Ensemble Optimal Inperpolation (EnOI), centre the observations and model on a single time and are
capable of resolving slowly evolving flows governed by simple balance relationships at synoptic scales. These methods have
provided useful states estimates and predictions, for example the National Centers for Environmental Prediction (NCEP's)
operational Numerical Weather Prediction (NWP) has used 3D-Var since the 1980s. Additionally, EnOI was effectively em-
ployed in Australia's Bluelink Ocean Data Assimilation System (Oke et al., 2008a). In Oke et al. (2010) a case was presented
for the use of EnOI, weighing up the predictive skill against its computational efficiency. Specifically, EnOI is highly compu-
tationally efficient as it does not represent the errors of the day; rather it assumes that the background error covariances are
well represented by a stationary or seasonally varying ensemble. More recent work has shown that combining flow-dependent
background error covariances (from an ensemble of model solutions) with a static ensemble achieves improved predictive skill
(Brassington et al., 2023).

With increasing computational capacity and the pursuit of more accurate weather and ocean forecasts over the last two
decades, a shift has been made to more advanced, time-dependent DA techniques (Edwards et al., 2015; Moore et al., 2019).
Advanced DA methods make use of the time-variable dynamics of the model allowing the observations to be assimilated over
a time interval given the temporal evolution of the circulation. In the atmosphere, these methods have provided considerable
improvement compared to the earlier, time-independent DA techniques, particularly for forecasts (e.g. Lorenc and Rawlins,
2005; Brousseau et al., 2012) and for highly intermittent flows with irregularly sampled observations (e.g. Xu, 2013). Indeed,
the two techniques that are the most promising in NWP are 4-Dimensional Variational Data Assimilation (4D-Var) and the
Ensemble Kalman Filter (EnKF), and ocean DA is following suit (Moore et al., 2019).

In 4D-Var the model and observations are combined using subsequent iterations of the tangent linear and adjoint models to
compute increments in the forecast model (initial conditions, boundary conditions and surface forcing) such that the difference
between the new model solution and the observations is minimised over a time window (Moore et al., 2004). With 4D-Var, a



continual and full estimate of the ocean over the assimilation window is created. This is ideal for both accuracy and timeli-
ness of current state estimates and future predictions, as a continuous field evolves by the nonlinear primitive equations. The
Kalman Filter (KF) can be formally posed in the same way as 4D-Var (Lorenc, 1986) and in practice uses an ensemble of
perturbed model simulations to approximate the model error covariances and their temporal evolution, and the ensemble mean
is considered the best estimate of the state of the system (Evensen, 2002). An advantage of generating an ensemble of forecasts
is that probabilistic forecasts can be derived from the ensemble spread.

Indeed, with the shift to more advanced DA techniques in ocean forecasting, it is important to quantify the improvements
gained. Here we use a Regional Ocean Modelling System (ROMS) configuration of a dynamic WBC (the EAC) to compare
two DA methods in a quantifiable manner. We compare the time-independent DA technique (EnOI) with the time-dependent
technique (4D-Var) using the same numerical model configuration and suite of observations. We quantify the differences in
predictive skill achieved by the two systems against assimilated and independent observations at the surface and subsurface.
We focus our analysis on the performance of the short-range (5-day) forecasts. After presenting the experiments (Section
2), we begin by comparing forecast performance against assimilated observations (Section 3.1). Then we employ a suite of
independent observations to assess the forecast skill of the two systems (Section 3.2). The model energetics (Section 4.2) and
the temporal and spatial scales of variability (Section 4.3) are then compared to understand what may drive differences in
predictive skill. Finally we summarise and discuss the way forward for improvements in Section 5.





**Figure 1.** (a) Mean Kinetic Energy from AVISO, with mean Eddy Kinetic Energy contours, showing the circulation in the EAC system and the model domain. The cyan lines show the sections through 278°S and 34°S. (b) Location of traditional observations used in the TRAD assimilation systems (SSH, SST, and SSS are not shown). (c) Location of additional observations used in the FULL assimilation system and for independent analysis herein. (d) Number of AVISO SSH and NAVO SST observations, and (e) number of Argo, XBT and SSS observations per 5-day assimilation window.



## 2 Model and Data Assimilation System Configuration

### 2.1 The Regional Ocean Modeling System Configuration

We use the Regional Ocean Modeling System (ROMS) to simulate the eddying ocean circulation off the southeastern coast of Australia between January 2012 and December 2013. This modelling suite is named the South East Australian Coastal Ocean Forecast System (SEA-COFS, Roughan and Kerry (2023b)). ROMS is a widely used free-surface, hydrostatic, terrain-following, primitive equation ocean model and is described by Haidvogel et al. (2000); Marchesiello and Middleton (2000); Shchepetkin and McWilliams (2005). The model configuration used in this study has been used in various past studies of the EAC and is described in detail in Kerry et al. (2016, 2020a); Roughan and Kerry (2023b).

The study domain covers SE Australia from 25.25°S to 41.55°S and approximately 1000 km offshore (Figure 1a). The domain covers the latitudinal extent of the EAC system from where the current jet is most coherent, the EAC separation region, the region of high eddy activity associated with the EAC eastern extension, and the EAC southern extension. The grid is rotated 20° clockwise such that the domain y-axis is oriented roughly parallel with the coastline. The cross-shore horizontal resolution varies from 2.5 km over the continental shelf and gradually increases to 6 km offshore. The horizontal resolution is 5 km in the along-shore direction. Higher resolution over the shelf allows the steep topography to be maintained while minimising pressure gradient errors that emerge in terrain-following coordinate schemes, which otherwise may result in artificial along-slope flow for steep topography (Haney, 1991; Mellor et al., 1994). As such, less topographic smoothing is required to ensure low horizontal pressure gradient errors while still representing the shelf and seamount structures in the model. The model utilises 30 vertical $s$-layers with higher resolution in the upper 500 m to resolve mesoscale dynamics and higher resolution near the seabed for improved representation of the bottom boundary layer. To better resolve surface currents, a near-constant-depth surface layer is provided by applying the vertical stretching scheme of De Souza et al. (2015).

Initial conditions and boundary forcing are derived from the Bluelink ReAnalysis version 3 (BRAN3; Oke et al., 2013). The boundary forcing is applied daily and misfits in baroclinic energy to the BRAN3 condition are absorbed at the boundary via a flow-relaxation scheme. The model is forced at the surface with realistic atmospheric forcing derived from the 12km resolution Bureau of Meteorology (BOM) Australian Community Climate and Earth-System Simulation (ACCESS) analysis (Puri et al., 2013). The atmospheric forcing fields are applied every 6 hours and used to compute the surface wind stress and surface net heat and freshwater fluxes using the bulk flux parameterisation of Fairall et al. (1996).

The free-running configuration, while unable to reproduce the temporal evolution of the mesoscale eddies, has been shown to accurately represent the mean dynamical features of the EAC and both the surface and subsurface (0-2000m) variability (Kerry and Roughan, 2020). Specifically, they show that the model accurately represents the mesoscale eddy related variability in SSH, the frequency in occurrence of EAC separation latitude, the seasonal cycle in SST, the ocean's subsurface structure based on data from Argo profiling floats, EAC transport and the temperature depth structure across the EAC. Thus, using data assimilation, we aim to constrain the model to reproduce the temporal evolution of the mesoscale eddies and examine the forecast skill achieved.



## 2.2 Observations

The same set of observations are assimilated into the ROMS model configuration using the two DA systems (EnOI and 4D-
Var) for comparison in this study. These include satellite-derived sea surface height (SSH), sea surface temperature (SST), sea
surface salinity (SSS), vertical profiles of temperature and salinity from profiling Argo floats and vertical profiles of temper-
ature from eXpendable Bathy-Thermographs (XBTs) (refer to Figure 1b). The number of processed observations assimilated
for each 5-day assimilation window is shown in Figure 1d,e. These observations are referred to as the "traditionally" available
observations (TRAD) (Siripatana et al., 2020). We describe the observations used and the observation uncertainties speci-
fied below. For a detailed description of the observations, the processing performed prior to assimilation, and the prescribed
observation uncertainties, the reader is referred to Kerry et al. (2016).

### 2.2.1 Satellite-derived Sea Surface Height

Archiving, Validation and Interpretation of Satellite Oceanographic Data (AVISO), France, produce global, daily, gridded
($1/4°$ x $1/4°$) mean sea level anomaly (SLA) data by merging of all available along-track satellite altimetry data, computed
with respect to a seven-year mean. We add the AVISO SLA data to the dynamic SSH mean from a long free run such that the
sea level data is consistent with the ROMS model configuration. The AVISO delayed-time global SLA product error for the
region is estimated at 2cm (CNES, 2015). We prescribe an additional 4cm of uncertainty to account for imbalances between this
statistical field and a dynamically-balanced SSH field required by the model, and the higher spatial-scale processes resolved by
the model compared to the gridded product. As such, we prescribe an observation uncertainty of 6cm. As the AVISO gridded
product poorly resolves continental shelf processes, we exclude SSH observations over water depths less than 1000m.

We use the gridded AVISO product to constrain SSH, rather than the along-track altimetry, for this comparison study. Current
work including the development of a high resolution coastal ocean forecast system (Roughan and Kerry, 2023a) is now making
use of along-track SSH data successfully with 4D-Var.

### 2.2.2 Satellite-derived Sea Surface Temperature

SST data from the US Naval Oceanographic Office's Global Area Coverage Advanced Very High Resolution Radiometer level-
2 product (NAVOCEANO's GAC AVHRR L2P SST) is used for this study. Data is available 2–3 times per day. We remove
day-time SST observations and any night-time observations when wind speed $< 2\ \mathrm{ms}^{-1}$ (Donlon et al., 2002). The percentage
of SST observations removed per 5 day cycle is 0.33-54.3 % (mean of 20.77%). As the resolution of the data is similar to the
resolution of the model, the observation uncertainty for the assimilation is chosen to be equal to the specified product error
(Andreu-Burillo et al., 2010) which is 0.4-0.5 °C.

### 2.2.3 Satellite-derived Sea Surface Salinity

We use the Level 3 gridded Sea Surface Salinity (SSS) product derived from the National Aeronautics and Space Administra-
tions's (NASA) Aquarius satellite (www.aquarius.umaine.edu/). This product provides daily fields at a $1°$ resolution. We set





the observation uncertainty to 0.4. The specified Aquarius SSS product error is ~0.2 and 0.4 is chosen to account for repre-

sentation errors. The value is considerably higher than the uncertainties specified for other *in-situ* salinity observations so SSS
provides little constraint to the system (Kerry et al., 2016, 2018).

### 2.2.4 Argo floats

Argo (free-drifting profiling) floats measure temperature and salinity of the upper 2000m of the global ocean (www.argo.ucsd.edu,
Figure 1b). The Argo data points are averaged to the model grid (in the horizontal and vertical) and a 5-minute time-step. Un-
certainty profiles are defined to specify the nominal minimal uncertainties for subsurface temperature and salinity (method
described in Kerry et al. (2016)). The profiles provide greater uncertainties in the depth ranges of greatest variability where
representation errors are likely to be the largest. The observation error variance is specified as the maximum of this nominal
minimum error variance and the variance of the observations from the same model cell.

### 2.2.5 eXpendable Bathy-Thermographs

eXpendable Bathy-Thermographs (XBT) collect temperature profiles along repeat lines sampled by merchant ships; the Sydney-
Wellington (PX34), and the Brisbane-Fiji (PX30) routes intersect our model domain (Figure 1b). Four PX30 lines and seven
PX34 lines took place over the assimilation period (2012–2013, Figure 1e). XBT casts are performed at 10 km intervals along
the sections and the XBT data points are averaged to the model grid and a 5-minute time-step. The nominal minimal uncer-
tainty variance profiles used for the Argo temperature observations are doubled for the XBT observations, and the observation
error variance is specified as the maximum of the nominal minimum error variance and the variance of the observations from
the same model cell.

### 2.2.6 Independent Observations used for system assessment

A suite of additional observations were also available over the simulation period (2012–2013) that were collected as part of
Australia's Integrated Marine Observing System (IMOS). These include surface velocity measurements from high-frequency
coastal radar (HF radar), temperature, salinity and velocity observations from continental shelf moorings off the coast of New
South Wales (NSW) and Southeast Queensland (SEQ), temperature, salinity and velocity observations from 5 deep water
moorings across the core of the EAC at 28°S (EAC array), and temperature and salinity observations from ocean gliders (refer
to Figure 1c). These products provide independent observations against which we assess the performance of the two systems.
Furthermore, these observations were assimilated into the ROMS model (along with the TRAD observations) using 4D-Var
(Kerry et al., 2016, 2018; Siripatana et al., 2020). Given the full suite of available observations were assimilated, this system
is referred to as the FULL system, and considered the 'best estimate' of the ocean state over the 2012-2013 period. As such,
the FULL system is also used in this paper as a benchmark against which to compare the performance of the two systems
presented in this study (4D-Var and EnOI systems that assimilate TRAD observations).





## 2.3 Data Assimilation Experiments

In this paper, we refer to three different configurations of the SEA-COFS model which differ in DA type and/or the observations assimilated. Each case is performed over the 2-year period from January 2012 and December 2013 and is described below.

1. 4D-Var TRAD: This refers to the 4D-Var system that assimilates "traditionally" available observations (SSH, SST, SSS, Argo and XBT). This system is similar to the system described in Kerry et al. (2016) expect that it only assimilates the TRAD observations.

2. EnOI TRAD: This refers to the system that assimilates the same observations as the 4D-Var TRAD but using the EnOI DA method described in Section 2.4.1 below.

3. 4D-Var FULL: This refers to the 4D-Var system that assimilates all available observations (SSH, SST, SSS, Argo, XBT, HF radar, shelf and deep moorings and glider data). It is similar to the system described in detail in Kerry et al. (2016, 2020b, 2018).

A detailed comparison of the 4D-Var TRAD and the FULL systems was presented in Siripatana et al. (2020). The purpose of this paper is to compare the 4D-Var TRAD and the EnOI TRAD systems, in order to provide a comparison of the two DA schemes using a common suite of traditionally available observations. We introduce the 4D-Var FULL system as a benchmark when comparing against observations that are independent to the TRAD experiments in Section 3.2.

## 2.4 Data Assimilation Methods

The classic state estimation problem can be given by

$$\boldsymbol{X}^a = \boldsymbol{X}^f + \boldsymbol{K}(\boldsymbol{y} - \mathrm{H}(\boldsymbol{X}^f)), \tag{1}$$

where $\boldsymbol{X}$ is the state estimate; superscripts $f$ and $a$ refer to forecast and analysis, respectively; $\boldsymbol{K}$ is the Kalman gain; $\boldsymbol{y}$ is the observation vector; H is the linear observation operator that interpolates the background circulation to observation 215 points in space and time. The $\boldsymbol{y} - \mathrm{H}(\boldsymbol{X}^f)$ term is referred to as the innovation vector and describes the difference between the observations and the forecast model mapped to observation space. The difference in DA techniques lies in the formulation of $\boldsymbol{K}$ which determines how the forecast innovations are mapped into model space to produce the new state estimate ($\boldsymbol{X}^a$). For the standard analysis equation that is solved by the Kalman Filter and the dual form of 4D-Var, $\boldsymbol{K}$ can be expressed as:

$$\boldsymbol{K} = \boldsymbol{B}\boldsymbol{G}^T(\boldsymbol{G}\boldsymbol{B}\boldsymbol{G}^T + \boldsymbol{R})^{-1}, \tag{2}$$

where $\boldsymbol{B}$ is the background covariance and $\boldsymbol{R}$ is the observation error covariance. $\boldsymbol{G} = \mathrm{H}\boldsymbol{M}_f$. For time-dependent methods, the $\boldsymbol{M}$ operator integrates over the model period using the model equations. This allows observations to be assimilated over a time





window and respects the dynamics of the model. In 4D-Var, $\boldsymbol{M}_f$ is the tangent-linear operator linearised about the forecast, while EnKF uses an ensemble of nonlinear model solutions. For 3D-Var and EnOI, this operation is replaced by the identity matrix as observations are all centered at a single time. Rather than using the model physics to constrain the model versus observation error, time- invariant covariances are prescribed.

### 2.4.1 EnOI

Ensemble methods (which include the time-dependent EnKF and the time-independent EnOI) use an ensemble of model anomalies to estimate the background error covariances. The EnKF allows for the time-varying statistics by using a fixed number of nonlinear model members (ensembles) to provide a statistical representation of $\boldsymbol{K}$. The ensembles are generated for every assimilation period so as to capture the state-dependent "errors of the day". For EnOI, the ensemble of model anomalies are generated from a long non-assimilating model run. This makes the assumption that the background error covariances are not state-dependent, and are well represented by a stationary or seasonally varying ensemble. This method is considerably less expensive than the time-dependent EnKF or 4D-Var methods as, once the stationary ensemble is generated, EnOI requires only a single integration of the nonlinear model to generate a background state, and only a single solution of the analysis equations to update the background. In constrast, to generate an analysis field using EnKF, the forward nonlinear model must be integrated $m$ times (where $m$ is the number of ensemble members) to represent the time-varying background error covariances and a background state (often based on the ensemble mean). All ensemble members are then updated, requiring $m$ solutions of the analysis equations. Therefore EnOI, is $m$ times less expensive than EnKF.

A challenge of ensemble methods is to determine the sufficient number of ensemble members to capture the entirety of the state-space, and techniques such as localisation and inflation are used to ensure unrealistic covariances are not applied (Houtekamer and Zhang, 2016). Specifically, localisation is used for three reasons: it reduces the fictitious large covariances at large distance due to sampling error; it improves the rank of the matrix inversion; and, by the use of a parametric form to taper to zero over the localisation distance the inversions become perfectly parallel improving computational efficiency (Gaspari and Cohn, 1999). Inflation is only applied to EnKF, not EnOI, with inflation of 5% being typical. The localisation and inflation techniques however remove some dynamical consistency from the solution. Recent work by the Australian BOM uses a hybrid-Ensemble Transform Kalman Filter (Sakov and Oke, 2008) based on 48-dynamic and 96-stationary ensemble members (Brassington et al., 2023). With EnOI, there is less constraint on the number of ensemble members, as the ensembles are only performed once to generate the stationary or seasonally-varying ensemble.

For EnOI, $\boldsymbol{M}$ in Equation 2 is set to the identity and all observations are colocated at a single time and the analysis equation (Equation 1) is considered only at that time. The background error covariance matrix is estimated from a static ensemble of model state anomalies and is given by:

$$\boldsymbol{B} = \frac{1}{m-1}\boldsymbol{A}\boldsymbol{A}^{T},$$ (3)

where $\boldsymbol{A}$ is the matrix of background ensemble anomalies; and $m$ is the ensemble size.



In the EnOI system used in this study, we use a stationary ensemble to represent the intraseasonal model anomalies. Each
member is calculated as a difference between a 2-week model average and a 2-day average, centered at the same time. This
is repeated every 30 days to ensure the anomalies are independent, generating 266 ensemble members. The DA system is run
with a 1-day cycle and centered observation window, so an analysis is generated every day. For SSH, temperature, and salinity,
the observation time is assumed to coincide with the analysis time, and innovations are calculated as the difference between
observation and model state at the analysis time. The localisation radius is set to 250 km for SSH, T, and S observations,
and to 100 km for SST observations. The observation errors are set equal to those described in Section 2.2 (identical for both
EnOI and 4D-Var systems), except for SST for which the error variance is increased by a factor of 2 for the EnOI system to
prevent overfitting to SST. The observation impact was moderated with an adaptive quality control procedure via the so-called
K-factor (Sandery and Sakov, 2017) with the value of K = 2.

For comparison with the 4D-Var system we perform 5-day forecasts based on the EnOI analyses every 4 days. Initial
conditions for each subsequent 5-day forecast are taken from the EnOI analysis. In this paper we focus on the forecast skill
between the 4D-Var and EnOI systems (not the analysis skill).

### 2.4.2 4D-Var

4D-Var uses variational calculus to solve for increments in model initial conditions, boundary conditions, and forcing such that
the differences between the observations and the new model trajectory is miminised – in a least-squares sense – over a specific
assimilation window. The goal is for the model to represent all of the observations in time and space using the physics of the
model, and accounting for the uncertainties in the observations and background model state, producing a description of the
ocean-state that is dynamically-balanced and a complete solution of the nonlinear model equations.

This is achieved by minimising an objective cost function, $J$, that measures normalised deviations of the modelled ocean
state (given the increment adjustments to model initial conditions, boundary conditions, and forcing) from the observations as
well as from the modelled background state (the model prior). The cost function is a function of the increment vector

$$\delta \mathbf{z} = (\delta \mathbf{x}(t_0)^T, \delta \mathbf{f}^T(t_1), ....., \delta \mathbf{f}^T(t_n), \delta \mathbf{b}^T(t_1), ....., \delta \mathbf{b}^T(t_n))^T \tag{4}$$

representing the increments to the initial conditions (time $t_0$), and the surface forcing and boundary conditions for model times
$t_1$ to $t_n$. The cost function can then be written as

$$J(\delta \mathbf{z}) = \frac{1}{2} \sum_{i=0}^{n} (\mathbf{H}_i \mathbf{M}(t_i, t_0) \delta \mathbf{z} - \mathbf{d}_i)^T \mathbf{R}_i^{-1} (\mathbf{H}_i \mathbf{M}(t_i, t_0) \delta \mathbf{z} - \mathbf{d}_i) + \frac{1}{2} (\delta \mathbf{z})^T \mathbf{B}^{-1} (\delta \mathbf{z}) = J_o + J_b \tag{5}$$

where $\mathbf{M}(t_i, t_0)$ represents the tangent linear version of the nonlinear model equations $\mathcal{M}$, integrated from $t_0$ to $t_i$. The
difference between the modelled background state and the observations is represented by the innovation vector, given at each
time $t_i$ by $\mathbf{d}_i = \mathbf{y}_i - \mathbf{H}_i(\mathbf{x}^b(t_i))$; where $\mathbf{y}$ are the observations and $\mathbf{H}_i$ is the linear operator that interpolates the background
circulation to observation points in space and time. $\mathbf{R}$ is the observation error covariance matrix and $\mathbf{P}$ is the background error
covariance matrix.



We seek to minimise the cost function by equating the gradient to zero. The gradient of the cost function is given by

$$\nabla_{\delta z} J = \sum_{i=0}^{n} \mathbf{M}(t_i, t_0)^T \mathbf{H}_i^T \boldsymbol{R}_i^{-1} (\mathbf{H}_i \mathbf{M}(t_i, t_0) \delta \mathbf{z} - \mathbf{d}_i) + \boldsymbol{P}^{-1} (\delta \mathbf{z}), \tag{6}$$

where $\mathbf{M}(t_i, t_0)^T$ is the adjoint of the tangent linear model equations. The desired analysis increment, $\delta z_a$, that minimises Equation 5 corresponds to the solution of equation $\nabla_{\delta z} J = 0$ and is given by

$$\delta z_a = (\boldsymbol{B}^{-1} + \boldsymbol{G}^T \boldsymbol{R}^{-1} \boldsymbol{G})^{-1} \boldsymbol{G}^T \boldsymbol{R}^{-1} \boldsymbol{d} \tag{7}$$

for the primal form (in model space) or

$$\delta z_a = \boldsymbol{B} \boldsymbol{G}^T (\boldsymbol{G} \boldsymbol{B} \boldsymbol{G}^T + \boldsymbol{R})^{-1} \boldsymbol{d} \tag{8}$$

for the dual form (in observation space). That is, the analysis increment corresponds to the right hand side of Equation 1, where $\boldsymbol{K}$ is given by Equation 2. For 4D-Var, the $\boldsymbol{M}_f$ in Equation 2 is the tangent-linear operator linearised about the forecast. For the transpose, $\boldsymbol{G}^T = \boldsymbol{M}_f^T \boldsymbol{H}^T$, $\boldsymbol{H}^T$ maps from observation to model-space and $\boldsymbol{M}_f^T$ represents the adjoint model operation

that integrates backward in time over the period.

In practice, with 4D-Var, subsequent integrations of the adjoint and tangent linear models are performed to solve for an increment vector that minimises (or acceptably reduces) $J$. To compute the gradient, the tangent linear model is integrated using the increment $\delta \mathbf{z}$ (for the first iteration, $\delta \mathbf{z} = 0$) and $\mathbf{H}_i \mathbf{M}(t_i, t_0) \delta \mathbf{z} - \mathbf{d}_i$ is computed. The adjoint model then computes the first term of Equation 6 and $\nabla_{\delta x} J$ is computed. A new increment, $\delta \mathbf{z}$, that reduces $J$ is generated using a Lanczos-based

conjugate gradient method. Subsequent increments to minimise $J$ are generated through subsequent interactions of the tangent linear and adjoint model (referred to as the *inner* loops).

After the last *inner* loop, the final increment is applied to the initial conditions, boundary and surface forcing and the new integration of the nonlinear model is performed. The integration of the nonlinear model given the increment adjustments that were solved for in the *inner* loops is referred to as the *outer* loop. The analysis field is given by the final integration of the

nonlinear model (the final *outer* loop) which provides a model state-estimate that is constrained to satisfy the nonlinear model equations (strong-constraint) and better represent the observations over the assimilation window. The analysis provides an improved estimate of the initial conditions for the next assimilation window. In this study we find that 15 *inner* loops and a single *outer* loop give an acceptable reduction in $J$ (rather than a true minimum).

To solve for the nonlinear ocean solution that better represents the observations, we must take into account the uncertainties

in the system. As such, the background (prior model) error covariance matrix, $\boldsymbol{P}$, and the observation error covariance matrix, $\boldsymbol{R}$, are important scaling factors in the cost function, $J$ (Equation 5). The background error covariance matrix should represent the expected uncertainties in the model initial conditions, surface and boundary forcings. We estimate $\boldsymbol{P}$ by factorisation, as described in Weaver and Courtier (2001), such that,





$$P = \mathbf{K}_b \Sigma \Lambda L_v^{1/2} L_h L_v^{1/2} \Lambda \Sigma \mathbf{K}_b^T, \tag{9}$$

where $\mathbf{K}_b$ are the covariance operators of the balanced dynamics, $\Sigma$ and $\Lambda$ are the diagonal matrices of the background error standard deviations and normalisation factors respectively, and $L_v$ and $L_h$ are the univariate correlations in the vertical and horizontal directions. We prescribe univariate covariance in $\mathbf{K}_b$ such that the dynamics are coupled through the use of the tangent linear and adjoint models but not in the statistics of $P$. The correlation matrices, $L_v$ and $L_h$, and the normalisation factors, $\Lambda$, are computed as solutions to diffusion equations following Weaver and Courtier (2001). The characteristic length

scales chosen for $L_v$ and $L_h$ are assumed to be homogeneous and isotropic, and their choice is detailed in Kerry et al. (2016). The specification of the observation error covariances is described in Section 2.2 above, and in more detail in Kerry et al. (2016).

Because we use the linearised model equations, the assimilation window length is limited by the time over which the tangent linear assumption remains reasonable (although longer windows have been shown to produce useful results). For the 4D-Var

system presented in this study, we find that a 5-day assimilation window is reasonable. A 5-day analysis is generated every 4 days (that is, there is a one day overlap between the analyses). Initial conditions for the subsequent 5-day forecast are taken from day 4 of the previous analysis.

The ROMS 4D-Var formulation and implementation is well described by Moore et al. (2011c, a, b), and it has been used successfully in many applications (e.g. Di Lorenzo et al., 2007; Powell et al., 2008; Powell and Moore, 2008; Broquet et al.,

2009; Matthews et al., 2012; Zavala-Garay et al., 2012; Janeković et al., 2013; Souza et al., 2014; Kerry et al., 2016; Gwyther et al., 2022; Wilkin et al., 2022). This work adopts the same 4D-Var configuration as described in detail in Kerry et al. (2016).

### 2.4.3 System Comparison

For the 15 *inner* loops and single *outer* loop used in this study, the 4D-Var data assimilation process is approximately 54 times more expensive than a single free run. This is comparable to the expense of an EnKF using 54 ensembles. The advantage

of EnKF (over 4D-Var) is that the tangent linear and adjoint models are not required and all calculations are performed in nonlinear space. The drawback is under-dispersion of the ensemble and the loss of dynamic consistency introduced through localisation and inflation. With a 4D-Var system, the use of the adjoint model can provide useful insight into the sensitivity of the ocean state to prior changes in state variables or forcings (e.g. Powell et al., 2013; Kerry et al., 2022) and the direct quantificiation of observation impacts (e.g. Powell, 2017; Kerry et al., 2018). Future work aims to compare the EnKF and

4D-Var methods, and explore Hybrid Ensemble-4D-Var methods that captialise on the advantages of both (i.e. the dynamical interpolation properties of the adjoint used in 4D-Var, and the explicit flow dependent error covariances of the EnKF (Lorenc et al., 2015)). This paper sets a baseline for future work by first comparing the existing and commonly used EnOI method with the 4D-Var method, across a common modelling framework and observational network.





# 3  System Performance: Assessing Predictive Skill

## 345  3.1  Assimilated Observations

We begin by assessing the performance of the EnOI and 4D-Var systems relative to the observations that the systems assimilate. The 5-day model forecast is compared to the observations that become available over those 5 days (that is, they have not yet been assimilated) to quantitatively assess the performance of the model forecasts over time. Comparing forecasts against observations provides objective assessment of the system performance.

Table 1 presents the mean innovation (Mean Absolute Difference, MAD), innovation bias (Mean Difference, MD), and number of observations for the 2-year period. Both systems have an identical number of observations. Compared to the EnOI, the 4D-Var improves the SST forecast error from 0.42°C to 0.36°C; the SSH forecast error — from 10.3 cm to 8.3 cm; *in-situ* temperature — from 0.90°C to 0.71°C; *in-situ* salinity — from 0.079 PSU to 0.056 PSU; and SSS — from 0.214 PSU to 0.183 PSU. Overall, the improvement of the MAD for the 4D-Var over the EnOI is 9–21%. The percentage differences in forecast error between the two systems are less for the surface observations (SLA, SST, and SSS) compared to the *in-situ* observations, indicating that the advantages of 4D-Var extend through the water column. In WBC regions, the parent model displayed MADs between reanalysed and observed SST values on day 1 of each assimilation of 0.2–0.6°C and MADs of 6–12 cm for SSH (Chamberlain et al., 2021b).

**Table 1.** Summary of performance of the EnOI and 4D-Var systems. Obs num refers to the average number of observations per 5-day assimilation window.

|        |         | SLA   | SST               | TEMP   | SAL    | SSS    |
|--------|---------|-------|-------------------|--------|--------|--------|
| EnOI   | MAD     | 0.103 | 0.424             | 0.901  | 0.0791 | 0.214  |
|        | MD      | 0.037 | -0.045            | -0.637 | -0.048 | 0.0417 |
|        | PER MAD | 0.102 | 0.404             | 0.882  | 0.0789 | 0.213  |
| 4D-Var | MAD     | 0.083 | 0.356             | 0.709  | 0.0560 | 0.183  |
|        | MD      | 0.031 | -0.035            | -0.534 | -0.032 | 0.0359 |
|        | PER MAD | 0.081 | 0.340             | 0.701  | 0.0554 | 0.182  |
|        | Obs num | 4518  | $1.58 \times 10^4$ | 273    | 198    | 137    |

The performance of the two systems relative to SSH, SST and Argo observations is presented in more detail using the 360 Root-Mean-Square Difference (RMSD) between the model forecasts at the observation locations, and the observation values. Figure 2a,b shows the RMSD between the forecasts (4D-Var and EnOI, respectively) and observations for SSH across the model domain, averaged over the 2-year period. The EnOI forecasts display higher SSH errors across the model domain, with both systems showing higher errors in the eddy-dominated region compared to the rest of the domain. Figure 2c shows that the spatially-averaged RMSD between the forecast and the observations is consistently higher for the EnOI forecasts over the 365  2-year period.




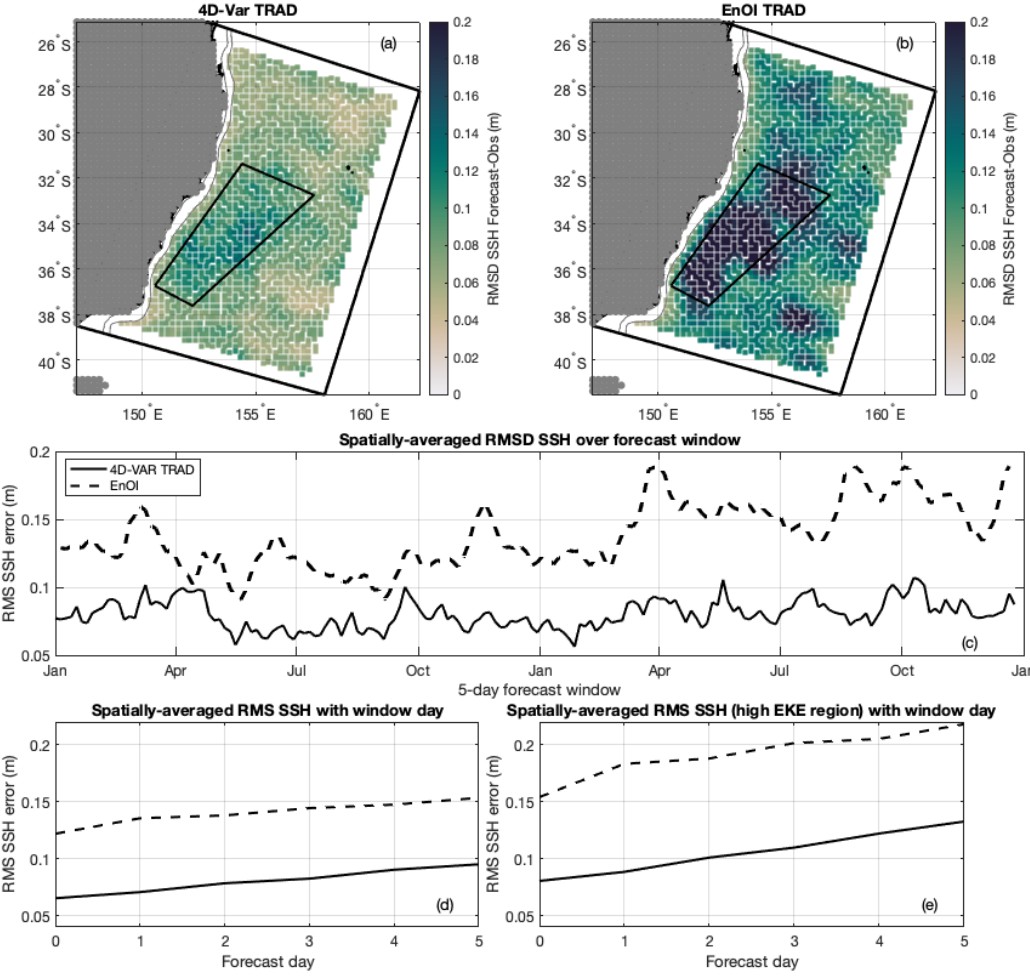

**Figure 2.** (a) RMSD between forecast and observed SSH for all 186 forecast cycles over the 2-year assimilation period for the TRAD 4D-Var system. (b) Same as (a) for the EnOI system. (c) Spatially-averaged RMSD between forecast and observed SSH for each 5-day forecast window. (d) Spatially-averaged RMSD between forecast and observed SSH for each day of the 5-day forecast window, averaged over the 186 forecast cycles. (e) Same as (d) but for the high EKE region (shown in (a) and (b)).

As each forecast is initialised from the previous analysis, forecast errors typically increase over the forecast horizon. SSH forecast errors are averaged across the model domain (Figure 2d) and for the eddy-dominated region (Figure 2e) for each day of the 5-day forecast horizon. With SSH, the forecast errors are consistently lower for the 4D-Var system due to lower errors in the initial conditions while the rate of error increase is similar between the 4D-Var and EnOI systems. At day 5, the domain-averaged (eddy-dominated region averaged) RMS SSH forecast errors are 61% (64%) higher for the EnOI system compared to the 4D-Var system.






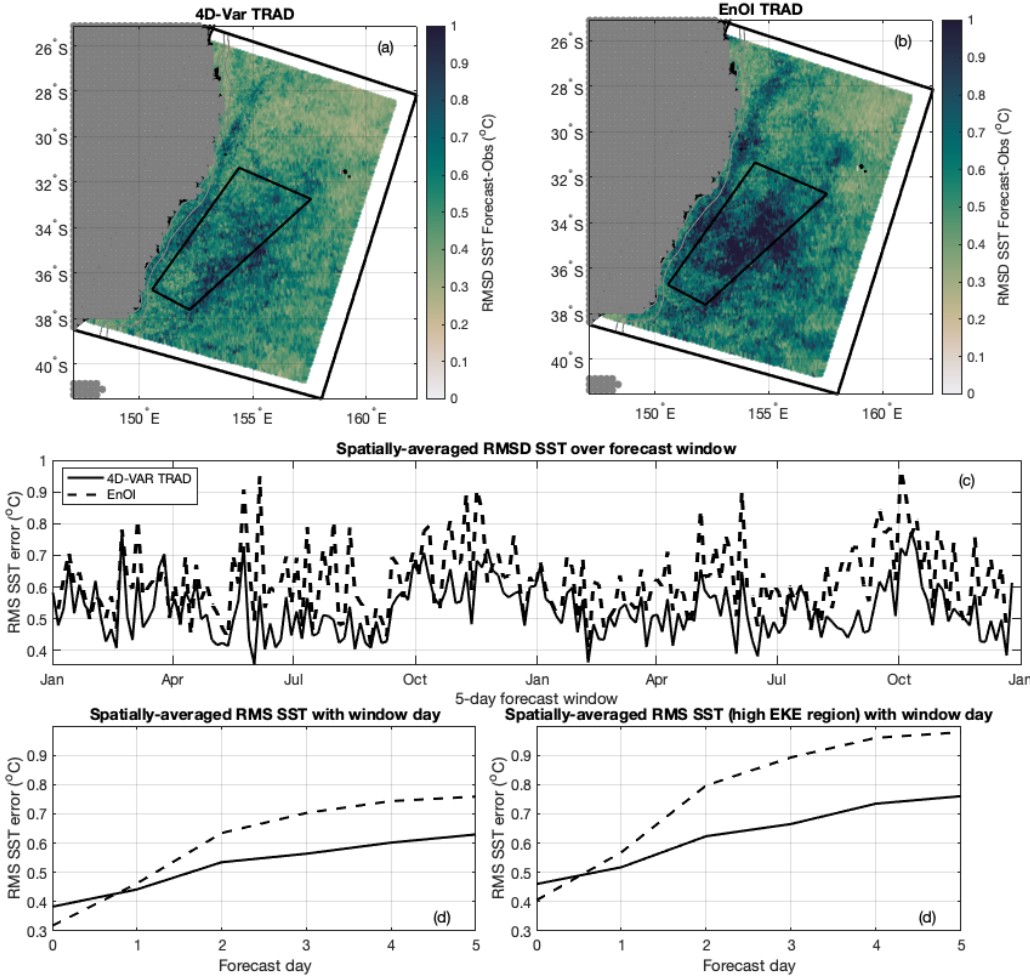

**Figure 3.** Same as Figure 2 but for SST observations.

In a similar manner to the SSH forecast errors in Figure 2, the forecast errors relative to SST observations are presented in Figure 3. Both systems display higher errors in the core of the EAC upstream of the typical separation region and in the eddy-dominated region. The EnOI forecasts display higher SST errors across the model domain, with the most pronounced difference in the eddy-dominated region (Figure 3a,b). The time series of RMSD for EnOI and 4D-Var (Figure 3c) are highly correlated as the statistics are sensitive to the number of observations and the coverage in the high variability area. While the EnOI analyses provide a slightly improved fit to SST (Figure 3d,e at day 0), SST forecast errors grow more quickly than in the 4D-Var system and the 4D-Var system outperforms the EnOI system for SST forecasts after 1-day. At day 5, the domain-averaged (eddy-dominated region averaged) RMS SST forecast errors are 21% (29%) higher for the EnOI system compared to the 4D-Var system.



To assess the subsurface predictive skill we extract the 5-day model forecast values at the observation times and locations for all Argo floats that observe in the region over the forecast window. Binning these observations with depth, we present profiles for temperature and salinity of the mean (Figure 4a,e), bias (Figure 4b,f), and the RMSD between the forecasts and the observations for all observations that fall on the first day of the forecasts (Figure 4c,g), and all observations that fall on day 5 of the forecasts (Figure 4d,h). The magnitude of the RMSDs can be compared to the Root-Mean-Squared (RMS) observation anomaly, which describes the variability of the observations within each depth bin. For *in-situ* temperature, both the 4D-Var and EnOI forecasts display similar skill on the first day of the forecasts (Figure 4c), however by day 5 the 4D-Var forecasts display lower errors compared to the EnOI forecasts over the upper 600 m, with a maximum difference in RMSD (bias corrected RMSD) of 0.56°C (0.34°C) at 200 m (Figure 4d). For salinity, forecast errors at day 5 are of similar magnitude throughout the water column for the two systems (Figure 4h). Both systems have RMS errors considerably less that the RMS observation anomaly. Salinity bias dominates the RMSD deeper than 600 m, so bias corrected RMSD values are less that the total RMSD (Figure 4g,h).

## 3.2 Independent Observations

As described in Section 2.2, a number of observations were withheld from the 4D-Var and EnOI DA systems presented in this paper, allowing the system performances to be assessed against independent observations. In this section, forecasts from the 4D-Var and EnOI systems (that assimilate the traditional suite of observations, TRAD) are compared to the analyses and forecasts produced by assimilating the full suite of observations (FULL). Comparisons are made between the observations and the model solutions extracted at the observation times and locations and predictive skill is assessed for days 1 to 5 of the forecast horizons (and analysis windows in the case of the FULL analysis).

Under the HF radar footprint at 30° S, surface radial velocity observations from two sources are combined to compute surface velocities to about 100 km offshore, covering the shelf and shelf slope circulation. This coverage typically includes the EAC as a coherent jet and the intermittent formation of cyclonic frontal eddies inshore of the EAC (Archer et al., 2017; Schaeffer et al., 2017; Kerry et al., 2020a). The complex correlations between the observed and model velocities are presented in Figure 5. At forecast day 5, the 4D-Var TRAD displays similar predictive skill to the FULL forecasts. The EnOI forecasts are worse than the 4D-Var TRAD across the 5 days, showing that the 4D-Var system provides better representation of the circulation under the HF radar footprint in the analyses and forecasts.

Glider data over the study period (2012–2013) was predominantly available over the NSW continental shelf in water depths <200 m, however, from May-July 2012, several glider missions extended offshore into eddies and sampled down to below 1000 m. These glider observations were shown to be particularly impactful in constraining transport and EKE estimates in the FULL simulation (Kerry et al., 2018). These observations represent independent data for the 4D-Var and EnOI TRAD systems, and Figure 6 shows how the simulations represent temperature and salinity as measured by the gliders.

Errors are lowest near the surface compared to over the thermocline region due to the assimilation of SST and SSS data in all three systems (4D-Var TRAD, EnOI TRAD and 4D-Var FULL). The 4D-Var TRAD has RMS forecast errors for temperature of a similar magnitude and depth structure as the RMS observation anomalies, and the errors do not considerably change from



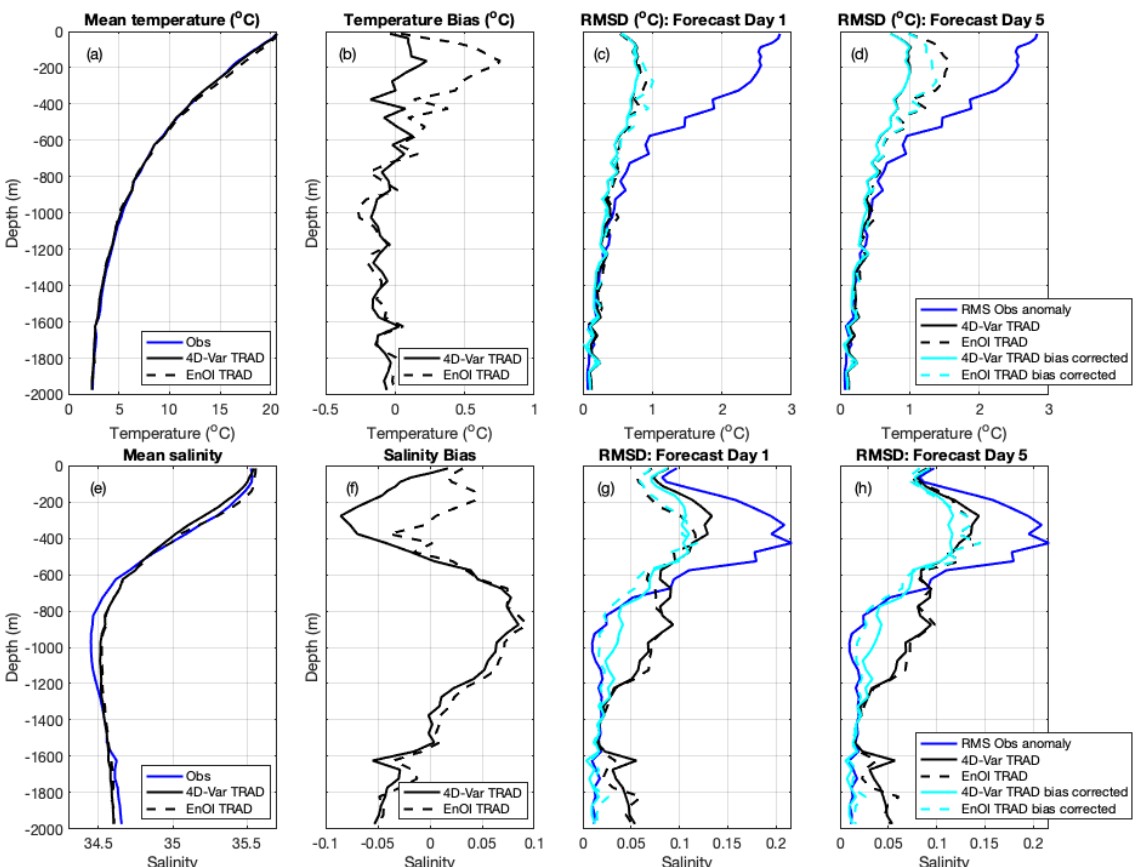

**Figure 4.** (a) Mean temperature observed by Argo floats and mean modelled temperature extracted at all Argo observation locations and times for 4D-Var and EnOI systems. (b) Temperature bias. (c) RMSD between forecast and observed temperature at all Argo observation locations and times that fall on forecast day 1, averaged over the 186 forecast cycles. (d) Same as (c) but for forecast day 5. (e-h) Same as (a-d) but for salinity observed by Argo floats.



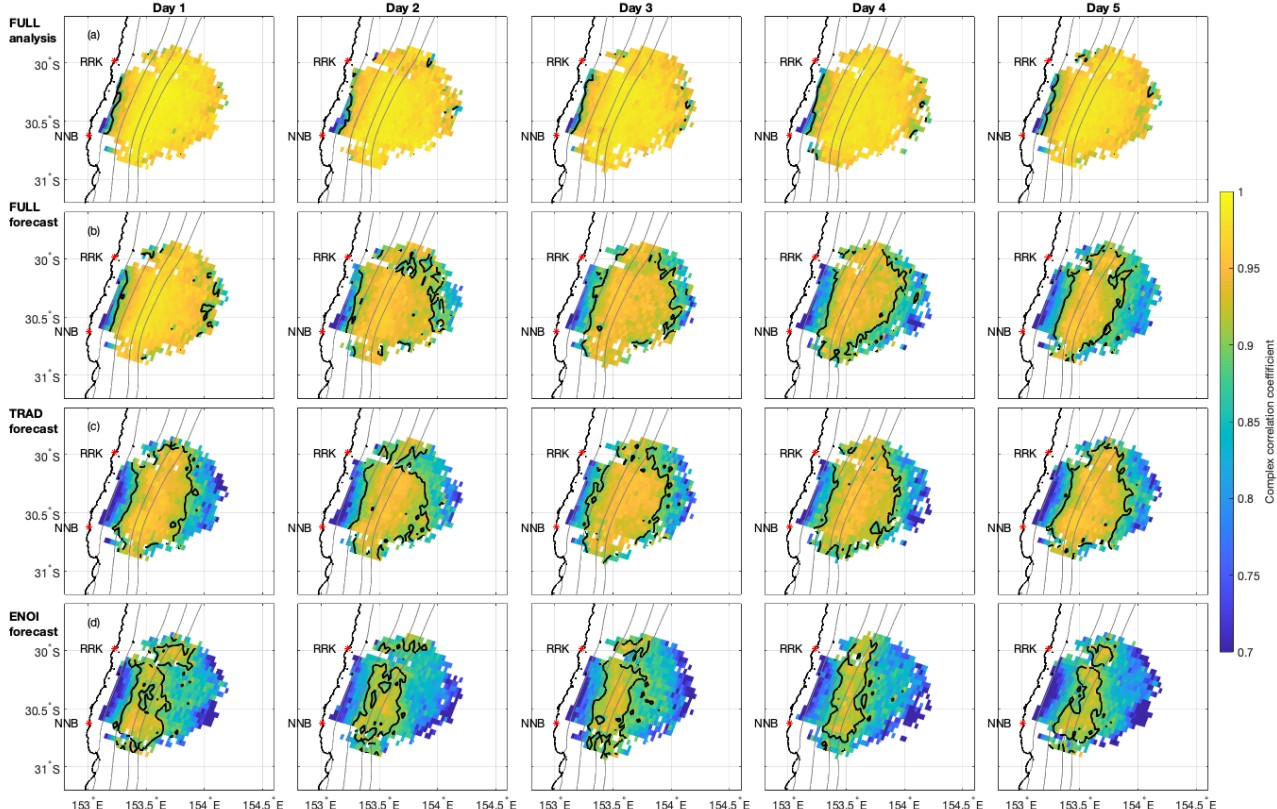

**Figure 5.** Complex correlation of daily-averaged surface velocities measured by the HF radar with FULL analysis (row a), FULL forecast (row b), TRAD forecast (row c) and EnOI forecast (row d), separated by window day (columns). Black lines show 0.9 complex correlation contour and gray lines show the 70, 200, 1000, and 2000 m isobaths. Only grid cells with a minimum of 15 velocity values over the 2-year period are shown, the values inside the 50 m isobath are removed as the computed velocities are unreliable here due to Geometric Dilution of Precision.

day 1 to day 5 of the forecast window. The EnOI errors are of similar magnitude to the 4D-Var near the surface ($\sim 1^{o}$C), but are 20% greater between 100-200 m for day 1 and 40% greater for that depth range at day 5 (Figure 6c,d). Temperature bias plays a considerable part in the EnOI RMSD values below 100 m, but the bias corrected RMSD for EnOI still exceeds the bias corrected RMSD for 4D-Var TRAD at both day 1 and day 5 (Figure 6c,d).

     For salinity, the 4D-Var and EnOI display similar forecast errors in the upper 200 m. This depth range corresponds to where
the many shelf glider observations exist. Below 200 m (the off-shelf missions into the Tasman Sea), forecast errors peak at 300 m reaching 0.30 for EnOI at day 5, compared to 0.23 for 4D-Var. Similar to the Argo observed salinity (Figure 4f,g,h), salinity bias dominates the errors associated with glider observed salinity below 500 m for 4D-Var TRAD and below 200 m for EnOI.



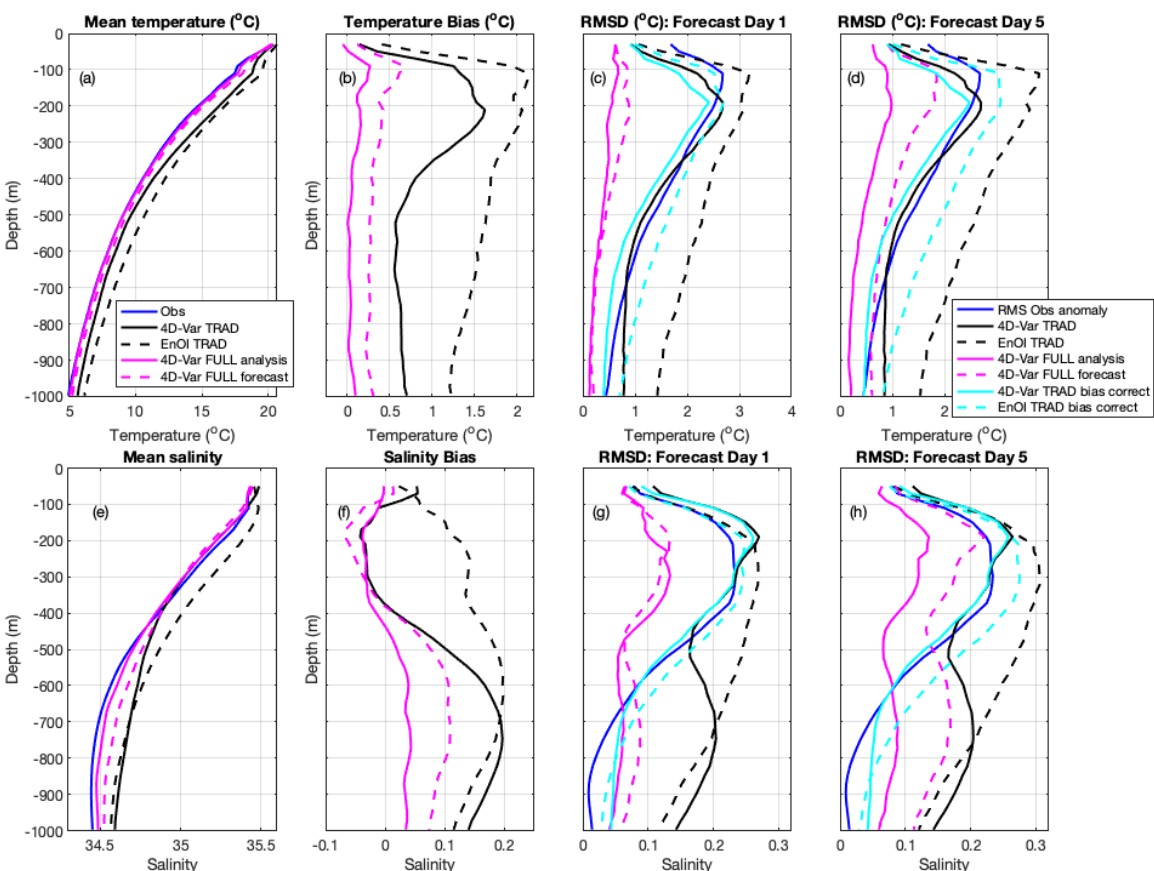

**Figure 6.** (a) Mean temperature observed by Gliders and mean modelled temperature extracted at all Glider observation locations and times for 4D-Var and EnOI systems. (b) Temperature bias. (c) RMSD between forecast and observed temperature at all Glider observation locations and times that fall on forecast day 1, averaged over the 186 forecast cycles. (d) Same as (c) but for forecast day 5. (e-h) Same as (a-d) but for salinity observed by Gliders.





Subsurface velocities are measured by ADCPs mounted on moorings in the EAC array, the SEQ shelf and slope, and on
the NSW shelf (Figure 1c). In Figure 7 we present the complex correlation between the modelled and observed velocities for
selected moorings extending from 28°S to 34°S. The mooring locations are shown on Figure 1c, with EAC2 and SEQ400
being in 1500m m and 400 m water depth at 28°S, CH100 being in 100 m water depth at 30°S and SYD100 being in 100 m
water depth at 34°S. At EAC2 and SEQ400, the 4D-Var TRAD display similar predictive skill to the FULL after 5 days and
considerably outperforms the EnOI system throughout the water column. This indicates the benefit of 4D-Var including the
northern boundary conditions in the cost function. On the shelf at 30°S (CH100) and 34°S (SYD100) the EnOI and 4D-Var
systems shown similar predictive skill.

As shown in both Figure 5 and Figure 7, the 4D-Var FULL complex correlations display a rapid reduction in correlation
by day 3-5 of the forecast. As discussed in Siripatana et al. (2020), while the analysis fits the velocity observations along the
continental shelf, the forecast model is unable to resolve the complexities of the shelf circulation such as the cyclonic vorticity
inshore of the EAC. As such, the forecast skill of the TRAD system is similar to that of the FULL system for 5-day forecast
horizons.

We have shown that the 4D-Var TRAD system outperforms the EnOI TRAD system at the surface and subsurface when
compared against both assimilated and independent observations. Improvements to temperature forecasts with 4D-Var are
more pronounced in the subsurface (the upper $\sim 400$ m) compared to at the surface (Figures 4 and 6). We now examine
the model forecasts to elucidate the differences between the representation of the ocean state (in model space, rather than
observation space) across the two DA systems.

## 4 Comparisons in Model Space

### 4.1 Initial Condition Increments

The model forecast, $\boldsymbol{X}_f$, is adjusted by the assimilation of observations (as per Equation 1) to produce an analysis, $\boldsymbol{X}_a$. This
model state estimate should provide a better representation of the observations and provides updated (improved) initial condi-
tions for the subsequent model forecast. In the 4D-Var system used in this study we perform a 5-day forecast and a 5-analysis
every 4 days, such that the initial conditions for the subsequent forecast are taken from day 4 of the previous analysis. For the
EnOI system, an analysis is generated every day. For consistent comparison across the 2 systems, we take the analysis every
4 days as initial conditions and perform a 5-day forecast. In both cases there are discontinuities in the ocean state between
day 4 of the previous forecast, and the beginning of the subsequent forecast (which correspond to concurrent times). This is
illustrated in Figure 8i, which shows a time series of temperature at the surface at $34^o$S. Assimilated (SST) and independent
(SYD140 mooring near-surface temperature data) are shown for reference. The discontinuities between the forecasts are less
pronounced for the 4D-Var system compared to the EnOI system. Over the entire 2-year test period, the RMSD between the
initial conditions (from the analysis) and the previous forecast field at that time illustrate greater discontinuities for the EnOI
system compared to the 4D-Var system for SSH, SST and subsurface temperature (Figure 8a-h).





**Figure 7.** Complex correlations between observed and modelled velocities for the 4D-Var TRAD forecast, the EnOI TRAD forecast, the FULL analysis and the FULL forecast, at selected mooring locations, separated by window days 1, 3 and 5 (columns). Each row represents a single mooring site. EAC2 (row a), SEQ400 (row b), CH100 (row c), and SYD140 (row d).



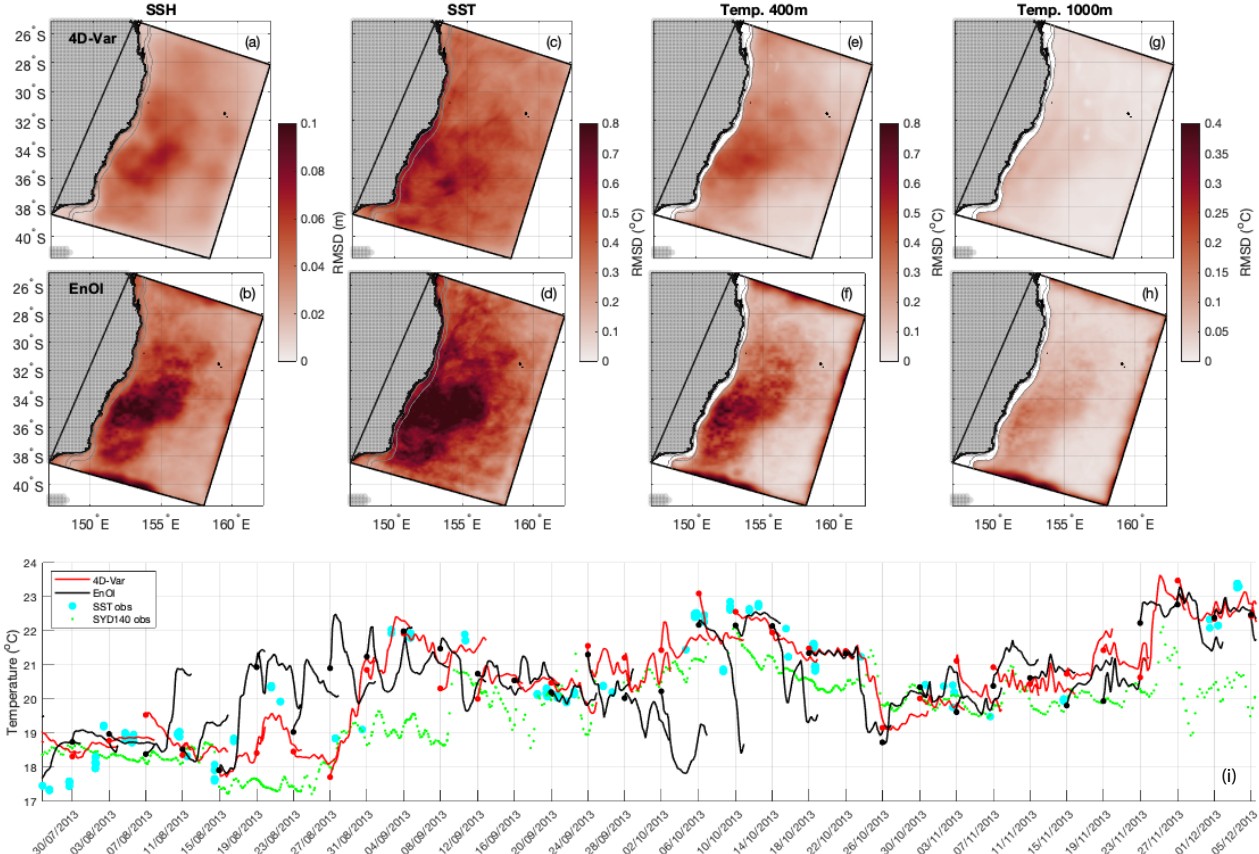

**Figure 8.** Root-Mean Squared Difference between the initial conditions (from the analysis) and the previous forecast field at that time for (a,b) SSH, (c,d) SST, (e,f) temperature at 400m and (g,h) temperature at 1000m, for 4D-Var system (top row) and EnOI system (bottom row). (i) Timeseries over an example period to illustrate the differences between the end of the forecast window and the analysis conditions in EnOI compared to 4D-Var, for surface temperature at $34°$ S (location shown in Figure 11l). SST observations within 2 grid cells, and temperature observations from SYD140 in the upper 25m, are also shown for comparison.

With 4D-Var we are able to represent the entirety of the observations collected over a time window (in this case 5 days), placing them in dynamical context using the (linearised) model equations. In contrast, EnOI performs discrete minimisations with observations centered on a single time (in this case every day). The estimate of the ocean over the observation window that is created with the 4D-Var assimilation system results is smaller discontinuities between forecast cycles, on average, compared to the EnOI system, as a continuous field evolves by the nonlinear primitive equations as opposed to starting a forecast from a discrete estimate, which can 'shock' the system. Our results of the improved predictability achieved by the 4D-Var system support the understanding that a continual and dynamically-balanced analysis field is advantageous to the quality of future predictions.





## 4.2 Energetics

The modelled velocities are used to compute eddy kinetic energy (EKE) and mean kinetic energy (MKE) over the 2012–2013 simulation period. MKE is given by $MKE = \frac{1}{2}(\overline{U}^2 + \overline{V}^2)$, where $\overline{U}$ and $\overline{V}$ are the time mean velocity components, and the EKE is given by $EKE = \frac{1}{2}(U'^2 + V'^2)$, where $U'$ and $V'$ are the velocity anomalies. The MKE describes the energy associated with the mean currents and the EKE describes the energy associated with the perturbations from the mean. Figure 9 shows the MKE and EKE averaged over the upper 400 m, and from 400-1200 m.

Comparisons of MKE above 400 m show that the EAC core is narrower and more confined to the slope in the 4D-Var system, while MKE for the EnOI system is more spread out and with higher MKE directly over the continental shelf (Figure 9a,e,i). This difference is despite the identical SSH observations being assimilated, noting that SSH observations in water depth <1000 m are not assimilated, and the identical forward numerical model. In the 4D-Var simulation, the MKE is greater below 400m than the EnOI simulation downstream of 27.5°S to the typical EAC separation zone (Figure 9b,f,j). This is consistent

with Kerry and Roughan (2020) who use a long-term integration of the free-running simulation to describe a downstream deepening of the EAC before separation.

The spatial structure of the EKE is similar across the two systems. Above 400 m, the EnOI system has elevated EKE over the EAC jet (Figure 9k, blue regions), while the 4D-Var system has elevated EKE in the eddy-dominated regions (Figure 9k, red regions). The elevated EKE for the EnOI system (in the more coherent region) relates to the greater discontinuities between the

subsequent forecasts, which manifests itself as greater low frequency >1 day variability over the 5 day forecasts as the 5-day model run adjusts to the "shocks" to the system. In contrast, the elevated EKE in the 4D-Var system outside of the coherent jet relates to the greater near-inertial variability. This is explored in Section 4.3 and Figure 12. At depth (400-1200 m), EKE is elevated for EnOI compared to 4D-Var in the EAC southern extension.

Eddies can form through barotropic instability in the mean flow or baroclinic instability in the vertical density structure. It is

important for a model to correctly represent these instabilities, as they represent the pathways by which eddies are generated. Following Kang and Curchitser (2015), we calculate the barotropic conversion rate (KmKe) as

$$\text{KmKe} = \rho_0 \left[ \overline{U'U'}\frac{\partial \overline{U}}{\partial x} + \overline{U'V'}\frac{\partial \overline{U}}{\partial y} + \overline{V'U'}\frac{\partial \overline{V}}{\partial x} + \overline{V'V'}\frac{\partial \overline{V}}{\partial y} \right], \tag{10}$$

where $\rho_0 = 1025 \, \text{kg m}^{-3}$. The baroclinic conversion rate (PeKe), from eddy potential energy to EKE, is calculated as

$$\text{PeKe} = -g\overline{\rho'W'}, \tag{11}$$

where the acceleration due to gravity is $g = 9.81 \, \text{ms}^{-2}$, and $\rho'$ and $W'$ are the density and vertical velocity anomalies. KmKe and PeKe have been previously used to explore eddy generation rates in the EAC (e.g. Li et al., 2021, 2022; Gwyther et al., 2023).

Barotropic and baroclinic energy conversions are computed from the model forecast fields and averaged over the 2-year period (Figure 10). Both the 4D-Var and EnOI systems show similar magnitude and overall spatial structure of the barotropic

and baroclinic energy conversions, and similar partitioning between barotropic and baroclinic instabilities. The similarities are





**Figure 9.** The 4D-Var simulation (a) 0–400 m, (b) 400–1200 m MKE, and (c) 0–400 m, (d) 400–1200 m time-averaged EKE are shown. The EnOI (e) 0–400 m, (f) 400–1200 m MKE, and (g) 0–400 m, (h) 400–1200 m time-averaged EKE are shown. In (i-l), the difference in each respective field between the 4D-Var and EnOI simulations are shown, where a positive difference indicates more energy in the 4D-Var simulation.





**Figure 10.** For the 4D-Var simulation, the (a) barotropic (KmKe) and (b) baroclinic (PeKe) conversions are shown. The EnOI simulation (c) barotropic and (d) baroclinic conversion rates are shown. Conversion rates are calculated as the depth-mean conversion for each model column from the surface to 450 m. In (e), the zonally-averaged conversions are shown for both simulations. Averaging is performed in the across-shelf direction in a band extending approximately from the coast to ∼ 3° offshore, as indicated by dashed lines in panel a.

likely due to the common model and atmospheric forcing. The barotropic conversion (compare Figure 10a,c) represents instabilities in the depth-mean flow, which 4D-Var and EnOI represent similarly. The baroclinic conversion (compare Figure 10b-d) is also similar between the DA configurations in overall spatial structure and the zonally-integrated magnitudes (Figure 10e), although the EnOI baroclinic conversion rate contains more high-wavenumber spatial patterns, which likely relate to unbalanced

adjustments upon assimilation. This is further explored in Figure 13 and the associated discussion.



### 4.3 Temporal and Spatial Scales of variability

When observations are assimilated the goal is to provide an improved fit to observations while retaining a dynamically-consistent ocean state. Ideally, upon assimilation of observations, the frequency and wavenumber spectra of the ocean state would remain unchanged. If energy is introduced at different, 'artificial' scales, this may impact the forecast skill. By pre-
senting the temporal and spatial scales of variability of the forecast ocean state we can understand how the assimilation has changed the ocean's energy distribution and understand the differences in error growth across the two DA systems.

The subsurface structure of the model fields and their variability is shown in Figure 11. The 'blue' regions in Figure 11f,l show more temperature variability in the EnOI system compared to 4D-Var, while the 'red' shows more temperature variability in 4D-Var. The EnOI has more temperature variability near the surface (upper 200-500 m) and the increase is greater in the
eddy-dominated region (Figure 11f,l), where adjustments are greater (Figure 8d,f) compared to upstream region. For velocity variability, 4D-Var shows elevated variability ('red' in Figure 11r,x) almost everywhere except in the upper 250 m near the shelf at $34^o$S. When the velocity or temperature variability is greater in EnOI compared to 4D-Var, we find that it is associated with greater discontinuities between the subsequent forecasts, which manifests itself as increased variability at low frequencies (>1 day) over the 5 day forecasts. This occurs as the 5-day model run adjusts to the 'shock' to the system of discontinuous
initial conditions. The discontinuities also exist in 4D-Var, but are less pronounced (Figure 8). When variability is greater in the 4D-Var system it is in the near-inertial band. This is illustrated by the analysis displayed in Figure 12, where frequency spectra are presented for the locations shown by black asterixes in Figure 11f,l,r,x.

Frequency spectra of all 5-day forecast windows with the model output 4-hourly gives a frequency range from 1/5 days to 1/8 hours, with 15 points in frequency space due to the short time series (31 points). Near the surface, energy is elevated in
the EnOI system for periods between 1-3 days for both temperature and velocity. This greater low frequency variability in the EnOI is more pronounced in the temperature fields (compared to velocity) and more pronounced in the eddy-dominated region ($34^o$S compared to the more coherent region at $28^o$S). Surface velocity (but not temperature) and subsurface temperature and velocity display elevated energy in the 16-24 hour band for the 4D-Var system compared to EnOI (Figure 12), corresponding to the near-inertial band. This inertial energy is introduced through the assimilation adjustments which, due to the nature of
4D-Var, must satisfy the model equations. Increased near-inertial variability upon 4D-Var data assimilation was also shown in Matthews et al. (2012); Kerry and Powell (2022b). Matthews et al. (2012) found that the increased inertial energy had minimal impact on the mesoscale circulation. Using Observing System Simulation Experiments Kerry and Powell (2022b) showed that, while the 4D-Var system displayed elevated near-inertial variability (compared to their free running *Truth* simulation), near-inertial frequencies did not influence energy at other frequencies and predictability at both higher frequencies (in their case internal tides) and lower frequencies (associated with the mesoscale circulation) was good.

The spatial scales of the forecast ocean state can be represented by wavenumber spectra. Here we present cross-shore wave number spectra through sections at $28^o$S and $34^o$S (Figure 1a) for days 1 and 5 of the forecast (Figure 13). There is elevated kinetic energy at finer length scales at the beginning of the forecast windows on average for EnOI and this energy dissipates by day 5 of the forecast. Specifically, elevated kinetic energy exists in the EnOI initial states at length scales less than 100 km





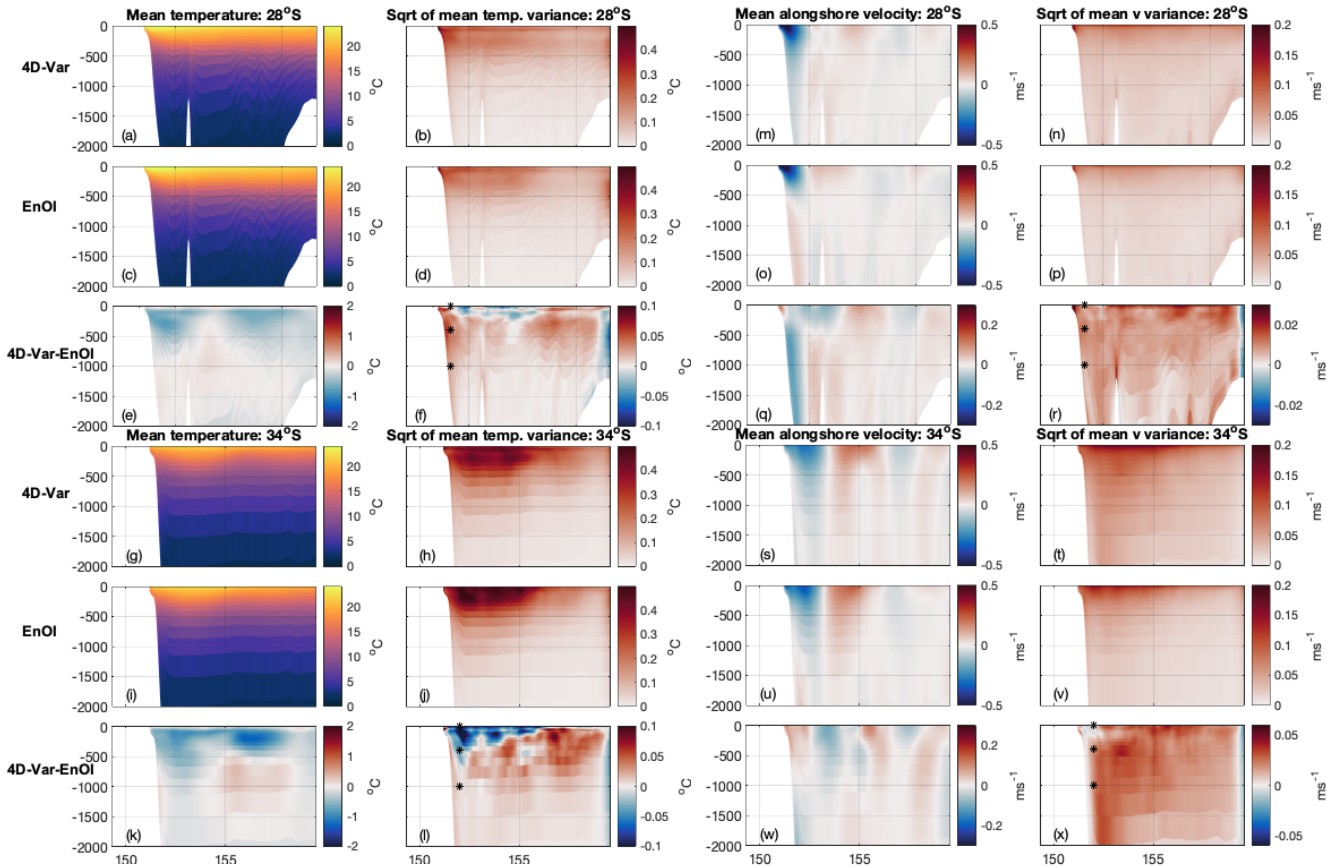

**Figure 11.** The mean temperature across all 5-day forecasts for section at 28°S for (a) the 4D-Var system, (c) the EnOI system, and (e) the difference in mean temperature between the systems (4D-Var - EnOI). Temperature variance is computed for every 5-day forecast, averaged over all forecast windows, and the square root taken. Shown for section at 28°S for (b) the 4D-Var system, (d) the EnOI system, and (f) the difference in mean temperature between the systems (4D-Var - EnOI). (g-l) shows the same as (a-f) but for section at 34°S. (m-x) shows the same as (a-l) but for alongshore velocity. Panels (l) and (x) shows points chosen to present frequency spectra (Figure 12). In the difference plots (f,l,r,x) red (blue) represents more (less) variance in the 4D-Var system compared to the EnOI system.



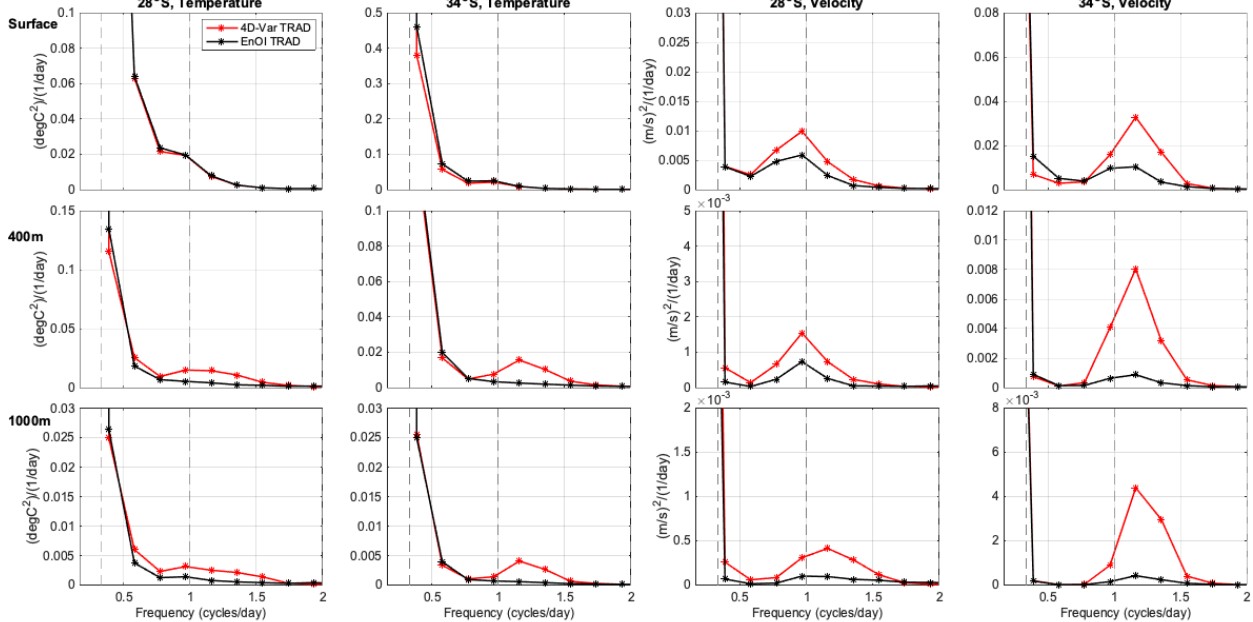

**Figure 12.** Frequency spectra in model space for temperature and alongshore velocity at the surface, 400 m and 1000 m at 28°S and 34°S. Spectra are computed for each 5-day forecast, then averaged. Points are chosen in the core of the EAC based on the long-term alongshore velocity mean (from Kerry and Roughan (2020)) where the shelf slope depth is 1500 m at 28°S and just offshore of the shelf slope where the water depth is 3500 m at 34°S. The 3-daily, daily and 12-hourly periods are shown by the vertical dashed lines.

at 28$^o$S and between 20-80 km at 34$^o$S. For the 4D-Var system the wavenumber kinetic energy spectra remains relatively unchanged upon assimilation, with the day 1 and day 5 wavenumber spectra tracking closely.

     The wavenumber kinetic energy spectra match the k$^{-5/3}$ slope for the mesoscale range, and the k$^{-3}$ slope for the submesoscale range (Figure 13) for the 4D-Var ocean state on day 1 and day 5, and the EnOI forecasts on day 5. The -5/3 and -3 spectral slopes imply surface quasi-geostophic and quais-geostrophic dynamics, respectively (Xu and Fu, 2011). For day 1

of the EnOI forecasts (representative of the analyses), energy remains elevated for the shorter length scales, and upon integration of the forecast model dissipates to match the slope associated with quais-geostrophic dynamics. This is in contrast to wavenumber kinetic energy analysis of the atmosphere by Skamarock (2004) who showed that the initial states of high-resolution NWP model forecasts lacked the fine-scale (mesoscale in the case of the atmosphere) energy because "observations to initialize the fine scales are not generally available and data assimilation methods that can use high-resolution observations

are not yet mature". Rather, the fine scale portion of the kinetic energy spectrum was spun-up in the forecasts in 6-12 hours. This forward energy cascade (from large to smaller scales) provides increased value to the NWP forecasts by allowing rapid spinup of mesoscale structure. In our study we observe an inverse energy cascade (from small to larger scales) over the EnOI 5-day forecast. While there is clear evidence of an inverse energy cascade in the ocean (e.g. Scott and Wang, 2005; Zedler



et al., 2019) at scales from the first baroclinic Rossby deformation radius to the basin scale, we find elevated kinetic energy at
scales less than 100 km due to EnOI's unbalanced adjustments resulting in an inverse cascade with associated degradation in
forecast skill.

The elevated energy at length scales less than 100 km occurs in the EnOI, despite the localisation scales being set to 250 km
for SSH, T, and S observations, and to 100 km for SST observations. For 4D-Var, the horizontal length scales of variability ($L_h$)
prescribed in the formulation of the background error covariance matrix (Equation 9) were set to 100 km for SSH, T and S.
The consistency of the wavenumber spectra upon assimilation in 4D-Var relates to the constraint that the analysis is a complete
solution of the model nonlinear equations.

## 5   Conclusions

This study shows in a quantified manner that the smoother and more dynamically-balanced fit between the observations and
the model's time-evolving flow achieved by the 4D-Var system results in improved predictability against both assimilated and
non-assimilated observations. The EnOI system does not produce as tight as fit to the SSH data as the 4D-Var system (although
this may be related to tune-able parameters in the DA formulation), however, the SSH error grows at the same rate in the EnOI
and 4D-Var forecasts (Figure 2). The surface expression of the EAC and its associated eddies is associated with the barotropic
mode, and our results show that the barotropic energy conversion rates are generally consistent across the two systems (Figure
10a,c). However, the baroclinic conversion rate has small spatial scale variability in the EnOI forecasts compared to the 4D-Var
(Figure 10b,d), and the EnOI analyses (the forecast initial conditions) display elevated energy at fine (<100km) spatial scales
(Figure 13). This is accompanied by reduced predictive skill for both surface and *in situ* temperature, *in situ* salinity and surface
velocities (Figures 3,4,5,6,7). For SST (Figure 3) and temperature in the upper 600 m (Figure 4c,d), the analyses have errors of
similar magnitude for the EnOI and 4D-Var systems, but error growth is considerably greater in the EnOI forecasts. Note that
the upper 600 m is the region of greatest variability in both temperature and salinity (Figure 4c,d,g,h, blue lines). The improved
forecasts of SST and *in situ* temperature in the upper 600 m for 4D-Var after 5 days (Figures 3, 4d) is a demonstration of
improved dynamical balance of the model initial conditions. This is evident by the smaller magnitude of the increments for
4D-Var (Figure 8a,c,e,g) compared to EnOI (Figure 8b,d,f,h). The bias corrected salinity errors also show similar errors at
forecast day 1 for both systems, with greater error growth in the EnOI system compared to 4D-Var by day 5 (Figure 4g,h).

Independent surface velocity observations as measured by the HF radar array at 30 $^o$S are less well represented by the EnOI
system compared to the 4D-Var system from day 1 through to day 5 of the forecasts (Figure 5). Independent *in-situ* temperature
observations from Gliders show only slightly lower analysis errors for 4D-Var compared to EnOI, but the subsurface temper-
ature forecasts degrade faster over the 5-day window for EnOI compared to 4D-Var (Figure 6), consistent with the forecast
errors associated with assimilated *in-situ* temperature observations (Figure 4). For salinity, EnOI and 4D-Var perform equally
well on the shelf (observations above 200m in Figure 6g,h are dominated by shelf gliders), but EnOI displays higher errors
below 200 m by day 5. 4D-Var displays improved velocity forecasts compared to EnOI for the upstream moorings (EAC2
and SEQ400, Fig 7), while downstream and on the shelf the forecasts are comparable. This indicates the benefit of 4D-Var



**Figure 13.** Cross-shore wavenumber kinetic energy spectra in model space at the surface, 400 m and 1000 m at 28°S and 34°S. The length scales 200 km, 100 km and 20 km are shown by the vertical dashed lines. The -5/3 and -3 spectral slopes are shown on the first panel for comparison. In computing the spectra, 2 ensembles and 2 bands are used to increase the statistical significance.



including the northern boundary conditions in the cost function. Generally, we show that the benefits of 4D-Var over EnOI are most pronounced in the (5-day) forecasts, rather than the fit of the analyses to the observations, consistent with Lorenc and Rawlins (2005)'s paper "Why does 4D-Var beat 3D-Var?".

The EnOI system displays greater discontinuities between the end of the forecast and the subsequent analysis, particularly for near-surface temperature (about the thermocline) and the discontinuities have greater magnitude in the downstream eddy-dominated region (Figure 8). These assimilation 'shocks' manifest as increased low frequency (> 1 day) variability (Figures 11 and 12). The 4D-Var system displays elevated energy in the near-inertial frequency band for both temperature and velocity (Figure 12), but maintains the kinetic energy distribution in wavenumber space (Figure 13). Consistent with Kerry and Powell

(2022a) and Matthews et al. (2012), the energy at near-inertial frequencies does not appear to affect the mean low frequency energetics associated with the mesoscale circulation.

    This study chose to compare two DA methods across a common modelling framework and observational network. The two methods were chosen as EnOI has been widely used by the Australian ocean forecasting community (Oke et al., 2008b, 2010; Chamberlain et al., 2021b), and 4D-Var has been implemented to study predictability and observation impact in the EAC (Kerry

et al., 2016, 2018; Siripatana et al., 2020; Gwyther et al., 2022, 2023). It made sense for the two user groups (operational and research) to come together to objectively compare the two methods. Each system was tuned by its developers (Australian Bureau of Meteorology for EnOI and Uni. NSW for 4D-Var). We note that the degree of fit between an analysis and the assimilated observations of a specific DA system is sensitive to the prior choice of various parameters, such as the observation and background error covariances, and that the system performance is influenced by the DA system configuration, such as size

of the ensemble (for ensemble methods) and the assimilation window length (for 4D-Var) (Moore et al., 2020; Santana et al., 2023). For example, the EnOI system presented here could be further tuned to provide an improved fit to SSH observations (Figure 2), and different ensemble sizes could be tested. For the 4D-Var system, different window lengths could be tested and the sensitivity to changes in $\boldsymbol{P}$ could be studied. However, the goal of this study was not to compare various versions of each DA method. Rather we compare a single version the two methods, carefully tuned by each user group, and set a baseline for

future comparisons. The focus of this paper is not the fit in the analyses, but the rate of forecast error growth and the response of the ocean state to the assimilation methodology. As such, the study's utility and relevance is significant without a large number of comparisons with different prior specified parameters or DA system configurations.

    The EnOI system is ~25 times cheaper than the 4D-Var system presented here. It is noted that EnOI has been effective for long-term reanalysis products where analyses were created every day (Oke et al., 2008b; Chamberlain et al., 2021b) and

forecasts were not required. With increasing computational capacity and the pursuit of more accurate ocean forecasts, this study's comparison motivates the use of 4D-Var over EnOI for ocean forecasts of the EAC region. This result is likely to be applicable over similar, highly variable, oceanic regions such as WBCs. More generally, the comparison advocates for the use of advanced time-dependent DA schemes over time-independent methods. We illustrate how a DA scheme can influence forecast skill which motivates future development of DA methods. It is noted that Australia's operational ocean model (OceanMAPS)

recently transitioned to an EnKF DA method (from EnOI). The new system achieves lower mean error and error variance in WBC extensions regions (Chamberlain et al., 2021a; Brassington et al., 2023), with lower increments to SSH and subsurface



velocities, and less kinetic energy at depth in the analyses, due to more dynamically-balanced adjustments, compared to the EnOI system.

Our future work specifically aims to directly address the need to improve predictive skill in WBC regions. Time-independent
schemes (e.g. 3D-Var and EnOI) are useful for intermittent cycling DA at synoptic scales, and are capable of resolving slowly evolving flows governed by simple balance relationships. Time-dependent DA methods (e.g. 4D-Var and EnKF) are greatly beneficial for highly intermittent flows with irregularly sampled observations as the time-variable dynamics of the model are used to evolve the error covariances. Furthermore, these methods allow the entirety of observations over a time interval to be minimised rather than discrete minimisations. The time-evolving state is required to truly exploit many novel observation types
that are nonlinearly or indirectly related to the model state. Indeed, the two techniques that are the most promising in NWP and ocean DA are 4D-Var and EnKF (Moore et al., 2019). In recent years it has been recognised that a marriage of 4D-Var and EnKF perhaps represents a more optimal approach since it capitalises on the advantages of both approaches (i.e. the dynamical interpolation properties of the adjoint, and the explicit flow-dependent error covariances that capture the "errors of the day"). The relative performance of 4D-Var and EnKF methods in regional ocean models has been assessed by Moore et al. (2020) and
the differences are due primarily to the properties of the background error covariances, so it is anticipated that the performance of a system using a hybrid covariance will be superior to either 4D-Var or the EnKF alone. Such ensemble-variational methods have been studied extensively for atmospheric DA (e.g. Lorenc et al., 2015) with improvements in forecast skill achieved particularly in dynamically active systems (Raynaud et al., 2011; Lorenc and Jardak, 2018).

*Data availability.* The ROMS model code is available from https://www.myroms.org.SEA-COFS model configuration is accessible at
https://doi.org/10.26190/5e683944e1369, https://doi.org/10.26190/5ebe1f389dd87, and https://doi.org/10.5281/zenodo.8294716.
The observations were sourced from the Integrated Marine Observing System (IMOS) – IMOS is a national collaborative research infrastructure, supported by the Australian Government. www.imos.org.au. Observations are available at www.aodn.org.au
Argo data were collected and made freely available by the International Argo Program and the national programs that contribute to it. (http://www.argo.ucsd.edu, http://argo.jcommops.org). The Argo Program is part of the Global Ocean Observing System (http://doi.org/10.17882/42182).
We acknowledge AVISO for the Delayed-time SLA data.The Ssalto/Duacs altimeter products were produced and distributed by the Copernicus Marine and Environment Monitoring Service (CMEMS) (http://www.marine.copernicus.eu).

*Author contributions.* CK developed the ROMS model configuration of the EAC system, processed the observations and developed the 4D-Var DA configuration. CK performed the 5-day forecasts given the EnOI analyses. CK analysed the results to produce Figures 1-8, 11-13. DG produced Figures 9-10. CK wrote the manuscript with some original input from AS. AS generated the results in Table 1. We acknowledge
Pavel Sakov who generated the EnOI analyses given the ROMS model configuration and the processed observations from CK. MR, SK, GB and JS provided useful guidance and input into the scope of the project and interpretation of results.



*Competing interests.* No competing interests are present.

*Acknowledgements.* This research, DG and AS were partially supported by the Australian Research Council Industry Linkage grant #LP170100498
to MR, CK and SK. Prior model development was supported by the Australian Research Council grants #DP140102337 and #LP160100162.
CSIRO Marine and Atmospheric Research and Wealth from Oceans Flagship Program, Hobart, Tasmania, Australia provided BRAN2020
output for boundary conditions.



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
