# Peer review of "Comparison of 4-Dimensional Variational and Ensemble Optimal Interpolation data assimilation systems using a Regional Ocean Modelling System (v3.4) configuration of the eddy-dominated East Australian Current System"

_EGUsphere, 2023_

## Author Comment (AC1)

This manuscript presents a very detailed and thorough comparison of two ocean data assimilation methods to East Australia Current (EAC): 4D-Var and EnOI. 4D-Var is a computationally intensive method that takes full advantage of the time-dependence of the circulation and ocean dynamics as described by the model, while EnOI is less computationally demanding, and uses information that is static in time. Both systems are currently employed in the Australia marine community, so a comparison of the two of considerable interest. I congratulate the authors on a very nice study which will be of interest to the broader oceanographic data assimilation community. I recommend publication after the authors have addressed my comments below, most of which are relatively minor.

The exceptions are section 2.4 and section 4.3.

**We thank the reviewer for the detailed response and feel that the suggestions have greatly improved the manuscript.**

Section 2.4 needs a bit of an overhaul since there is some repetition and the notation used throughout is not consistent. See below for more detailed comments.

**We have overhauled this section to ensure that all notation is correct and the data assimilation methodology is correctly described for the general case, and the specific 4D-Var and EnOI systems. Refer to specific comments below and the Tracked changes document.**

Section 4.3 is highly speculative and unconvincing for this reviewer.

**The robustness of the results presented in Section 4.3, and the associated discussion, has been improved by including comparisons to the Free-run and to observations. We have changed the discussion to remove any speculation and now ensure that the discussion and conclusions highlight only what the results explicitly show. Refer to updated Figures 11-14, the specific comments below and the Tracked changes document.**

Lines 64-66: The reference to 3DVar in NWP seems a bit out of place here. A better ocean reference here would be NEMOVAR run at ECMWF which uses 3D-Var.

**Thankyou. This sentence has been replaced with "the European Centre for Medium-Range Weather Forecasts uses 3D-Var to produce initial conditions for its coupled ocean-atmosphere modelling system (Mogensen et al., 2012)"**

Line 214: H(.) does not have to be linear. In 4D-Var, for example, it includes the nonlinear model.

**The word linear has been removed. Thank you.**

Line 214: Replace "interpolates" with "samples"

**The sentence now reads "H is the observation operator that samples the background circulation to observation points in space and time."**

Line 220: Your equation for **G** does not represent the general case. **G** is the tangent linearization of $H(.)$. The equation **G=H\*M** stated here implies a single observation time at the end of the forecast window. More generally, **G** would be given by the sum of terms involving H\*M at each of the observation time.

Line 221: Do not use bold font here since this implies that this is a matrix. The forecast model though will, in general, be non-linear so it cannot be represented by a matrix. Use the same font as you use at line 280.

Line 221: Replace "model" with "nonlinear model"

**This paragraph has been modified accordingly. Please refer to the tracked changes document.**

Line 249: This **M** should be "**M**$_f$"

**This was removed with the changes made above.**

Lines 259-260: Can you say a bit more about the localization operator - what localization function do you use, and where is it applied in the equivalent of equation (2)?

**We have included more detail on the localisation method. "The localisation method applied is based on local analysis (Ott et al., 2004); that is an analysis of a local region is produced with a local background error covariance matrix that has lower dimension that the full state vector. The local analyses are then used to construct complete model states for advancement to the next forecast time. Performing the data assimilation analysis locally is convenient for parallelising the solver. In addition to this, a polynomial taper function is applied to bring the covariance to exactly zero on a specified length scale (Gaspari and Cohn, 1999). The localisation radius is set to 250 km for SSH, T, and S observations, and to 100 km for SST observations."**

Equation (4): Use upper-case **X** to be consistent with equation (1).

**Changed.**

Equation (5): Replace **HM** by **G** to be consistent with equation (2).

**Done**

Line 280: The font you use here to represent the nonlinear model should be what you use at line 221.

**Addressed.**

Line 282: Insert "... introduced above" after the expression for **d**.

**Added.**

Line 282: The "H" operator used in your expression for **d** and later on this line should not be bold; it is the same as the nonlinear observation operator introduced at line 215.

**Addressed throughout.**

Line 282: Replace "interpolates" with "samples"

**Addressed.**

Line 282: Replace lower-case **x** with upper-case **X**. Also you introduce superscript "b" here while in equation (1) you use "f". Use a consistent notation throughout otherwise it looks like you are talking about different objects.

**Addressed.**

Line 283: Replace "**P**" with "**B**" (here and throughout) to be consistent with equation (2).

**Addressed**

Equation 6: Replace "**HM**" with "**G**" and "**P**" with "**B**"

**Addressed.**

Lines 289 and 290: Delete equation (7) and line 290 since they are irrelevant here. It is (8) that is consistent with the form of the Kalman filter gain matrix.

**Done.**

Equation (8): Use superscript "a" instead of subscript to be consistent with equation (1).

**Consistent subscripts everywhere now.**

Lines 293-295: Delete the last two full sentences -  this is repetitive information.

**Done.**

Line 298: Replace **HM** with **G.**

**Done.**

Line 298: The sentence beginning "The adjoint model then computes..." is true only for the primal form of 4D-Var which I understandyou are not using here.

**This sentence has been removed.**

Line 317: Delete "univariate covariance" and replace $\mathbf{K_b}$ with $\mathbf{K_b}=I$.

**Addressed.**

Line 318: Replace **P** with **B**.

**Addressed.**

Line 320: It would be helpful to include a table here that summarizes the correlation lengths assumed for the control vector elements.

**Added, new Table 1.**

Line 333: In the 4D-Var experiments, are you adjusting only the initial conditions, or are you adjusting the surface forcing and open boundary conditions as well? If the latter, this represents another significant difference between the 4D-Var strategy and the EnOI strategy.

**In generating the 4D-Var analysis, we adjust the initial conditions, boundary and surface forcing. This is now reiterated in the paragraph directly above. "We adjust the model initial conditions, boundary conditions and surface forcing such that the new model solution (the analysis) better represents the observations over the assimilation interval. Open boundary conditions are adjusted every 12 hours and surface forcing every 3 hours. "**

Line 334 and 335: Another advantage of the EnKF is that the ensembles members can be run simultaneously if sufficient computing resources are available.

**Thank you, added.**

Line 338 and 339: Observation impacts can be computed from ensemble methods also using ensemble FSOI e.g.: Liu, J., Kalnay, E., 2008. Estimating observation impact without adjoint model in an ensemble Kalman filter. Q. J. R. Meteorol. Soc. 134, 1327–1335.

**Added, thank you.**

Table 1: What does "PER MAD" refer to in the table? It is not mentioned anywhere in the main text. Remove from the table if is not relevant.

**We have removed this as we do not discuss it in the text.**

Lines 449 and 450: Do these discontinuities/differences correspond to the DA increments?

**What we present are not exactly analysis increments, hence the Section name 'Initial Condition Increments' rather than "Analysis Increments'. This is clarified in a new paragraph that reads:**

**"The discontinuities presented here do not exactly correspond to the analysis increments. We have presented the differences in the ocean state between day 4 of the previous (5-day) forecast, and the beginning of the subsequent forecast (which correspond to concurrent times). For 4D-Var, the ocean state at the beginning of the forecast is taken from the previous cycle analysis, and so the difference presented here represents the**

difference between the forecast (or the background) at day 4 and analysis at day 4 (once data assimilation has been performed on that assimilation cycle) . This is essentially the `analysis increment at day 4', however for a 4D-Var system the analysis increments typically refer to the adjustments to the initial conditions, boundary and surface forcing that are made to generate the analysis. For EnOI, the analysis increments refer to the difference between the background model and the analysis (both centred on a single time and computed daily in this case). However, here we take the analyses every 4 days and perform 5-day forecasts, and the differences presented here refer to the difference between day 4 of the forecast and the analysis that provides initial conditions for the subsequent forecast."

Lines 503 and 504: This statement is not necessarily always true. If the model background is deficient at some space- and/or time-scale, then these may be corrected by DA so that the analyses and forecasts are better.

We now acknowledge this and have rewritten the paragraph to read "When observations are assimilated the goal is to provide an improved fit to observations while retaining a dynamically consistent ocean state that can be used an initial conditions for the subsequent forecast. The background numerical model produces an estimate of the ocean state whose frequency and wavenumber spectra are limited by the resolution of the model and the processes resolved. If the observations sample time and space scales that cannot be resolved by the model, it is standard DA practice to either remove these scales of variability from the observations or account for them in the observation error terms (e.g. Kerry and Powell 2022). If the model background is deficient at some space- and/or time-scale (that it is able to resolve), then these may be corrected by DA so that the analyses and forecasts are better. However, if the assimilation process introduces energy at different, non-physical scales, this may negatively impact the forecast skill. By presenting the temporal and spatial scales of variability of the forecast ocean state we can understand how the assimilation has changed the ocean's energy distribution and understand the differences in error growth across the two DA systems."

Figure 11: It would be helpful to also show the mean and sqrt of the variance from a free run of the model without DA to see how assimilation changes these fields along the two section shown.

Comparisons with the Free-run along the two sections have been added to Figure 11. The Free-run has also been included in comparisons in both frequency and wavenumber space in Figures 12-14.

Lines 537 and 538: I think it is a stretch to think that you are adequately resolving the submesoscale here.

We agree, we only partially resolve the submesoscale. This has been noted in the text.

Lines 547 and 548: Did you actually calculate the spectral transfer function? The approximate slope of the wavenumber spectrum alone is not enough to infer that there is an inverse energy cascade. While dx=2.5 km over the shelf, the effective resolution of the

model is probably more like 3dx or 4dx, and off the shelf dx is larger. Various published studies show that there is a forward energy cascade at the ocean submesoscale when it is adequately resolved. There is also a suggestion that the canonical slope should be -2. The following is an excellent review article: McWilliams, J.C., 2016: Proc. R. Soc. A, 472, 20160117.

Line 550: After cascade insert "and consistent with". That said though, I am not convinced by the arguments you make here unless you demonstrate by direct calculation that there is in fact an inverse energy cascade in your model.

**We agree. We cannot imply an inverse energy cascade without computing the spectral transfer function. However we do see an introduction of energy at small spatial scales upon assimilation with EnOI, and that this elevated small scale energy is lost by day 5. We removed discussion of an inverse energy cascade and now state only exactly what our results show.**

Lines 531-556: I find this whole section to be quite speculative. The canonical spectral slopes for QG and SQG are derived from highly idealized, and unforced simulations. Numerous model studies with forcing, and enhanced beta-effect (i.e. bathmetry) indicate that other slopes are possible. In addition, are the arguments made here consistent with the barotropic and baroclinic conversions discussed earlier. Barotropic and baroclinic instabilities are fundamentally very different in nature, so it is not clear to what extent one would expect the canonical cases to apply in a mixture of the two.

**We now state only what is shown in Fig 14 with no speculation.**

**We clarify that we do not resolve the submesoscale and that the spectral slopes were only added to the figure for reference and to illustrate an approximate match, however our results do not rely on this match at all. We do not imply any different dynamics in the two DA systems based on the different slopes. We only aim to show the differing distribution of energy at various spatial scales. We also no longer imply an inverse energy cascade.**

**We have added wavenumber kinetic energy spectra from the Free-run, and from AVISO gridded geostrophic velocities (at the surface) which strengthens the robustness of the results and the discussion.**

Line 663: Replace **P** with **B**.

**Done**